# Intronic enhancer region governs transcript-specific *Bdnf* expression in rodent neurons

**Jürgen Tuvikene[1,2]\*, Eli-Eelika Esvald[1,2†], Annika Rähni[1†‡], Kaie Uustalu[1†], Anna Zhuravskaya[3], Annela Avarlaid[1], Eugene V Makeyev[3], Tõnis Timmusk[1,2]\***

[1]Department of Chemistry and Biotechnology, Tallinn University of Technology, Tallinn, Estonia; [2]Protobios LLC, Tallinn, Estonia; [3]Centre for Developmental Neurobiology, King's College London, London, United Kingdom

**Abstract** Brain-derived neurotrophic factor (BDNF) controls the survival, growth, and function of neurons both during the development and in the adult nervous system. *Bdnf* is transcribed from several distinct promoters generating transcripts with alternative 5' exons. *Bdnf* transcripts initiated at the first cluster of exons have been associated with the regulation of body weight and various aspects of social behavior, but the mechanisms driving the expression of these transcripts have remained poorly understood. Here, we identify an evolutionarily conserved intronic enhancer region inside the *Bdnf* gene that regulates both basal and stimulus-dependent expression of the *Bdnf* transcripts starting from the first cluster of 5' exons in mouse and rat neurons. We further uncover a functional E-box element in the enhancer region, linking the expression of *Bdnf* and various pro-neural basic helix–loop–helix transcription factors. Collectively, our results shed new light on the cell-type- and stimulus-specific regulation of the important neurotrophic factor BDNF.

**\*For correspondence:**
jurgen.tuvikene@taltech.ee (JT);
tonis.timmusk@taltech.ee (TT)

[†]These authors contributed equally to this work

**Present address:** [‡]Protobios LLC, Tallinn, Estonia

## Introduction

Brain-derived neurotrophic factor (BDNF) is a secreted protein of the neurotrophin family (*Park and Poo, 2013*). During the development, BDNF promotes the survival of various sensory neuron populations (*Ernfors et al., 1994*; *Jones et al., 1994*). In the adult organism, BDNF is also required for the proper maturation of synaptic connections and regulation of synaptic plasticity (*Korte et al., 1995*; *Park and Poo, 2013*). Defects in BDNF expression and signaling have been implicated in various neuropsychiatric and neurodegenerative diseases, including major depression, schizophrenia, Alzheimer's disease, and Huntington's disease (*Autry and Monteggia, 2012*; *Burbach et al., 2004*; *Jiang and Salton, 2013*; *Murray et al., 1994*; *Ray et al., 2014*; *Wong et al., 2010*; *Zuccato et al., 2001*; *Zuccato and Cattaneo, 2009*).

Rodent *Bdnf* gene contains eight independently regulated non-coding 5' exons (exons I–VIII) followed by a single protein-coding 3' exon (exon IX). Splicing of one of the alternative exons I–VIII with the constitutive exon IX gives rise to different *Bdnf* transcripts (*Aid et al., 2007*). Additionally, transcription can start from an intronic position upstream of the coding exon producing an unspliced 5' extended variant of the coding exon (exon IXa-containing transcript) (*Aid et al., 2007*). The usage of multiple promoters enables complex cell-type- and stimulus-specific *Bdnf* expression (reviewed in *West et al., 2014*). For instance, *Bdnf* exon I-, II-, and III-containing transcripts show mainly nervous system-specific expression patterns, whereas *Bdnf* exon IV- and VI-containing transcripts are expressed in both neural and non-neural tissues (*Aid et al., 2007*; *Timmusk et al., 1993*). Similar expression patterns for different *BDNF* transcripts are also observed in humans (*Pruunsild et al., 2007*). Notably, different *Bdnf* transcripts have distinct contribution to various aspects of neural

circuit functions and behavior (*Hallock et al., 2019*; *Hill et al., 2016*; *Maynard et al., 2016*; *Maynard et al., 2018*; *McAllan et al., 2018*; *Sakata et al., 2009*).

In addition to proximal promoter regions, the complex regulation of gene expression is often controlled by distal regulatory elements called enhancers (reviewed in *Buecker and Wysocka, 2012*). Enhancers are usually active in a tissue- and cell-type-specific manner (reviewed in *Heinz et al., 2015*; *Wu et al., 2014*), and can be located inside or outside, upstream or downstream of the target gene, within another gene or even on a different chromosome (*Banerji et al., 1981*; *Lettice et al., 2003*; reviewed in *Ong and Corces, 2011*). Many enhancers are activated only after specific stimuli, which cause enrichment of active enhancer-associated histone modifications and increased chromatin accessibility (*Su et al., 2017*). Genome-wide analysis has proposed approximately 12,000 neuronal activity-regulated enhancers in cortical neurons (*Kim et al., 2010*). Importantly, dysregulation of enhancers or mutations in proteins that participate in the formation of enhancer-promoter complexes is associated with a variety of disorders, including neurodegenerative diseases (reviewed in *Carullo and Day, 2019*).

Previous studies from our laboratory have suggested that the expression of the *Bdnf* gene is also regulated via distal regulatory regions. Notably, the induction of *Bdnf* mRNA after BDNF-TrkB signaling in neurons seems to depend on unknown distal regulatory regions (*Esvald et al., 2020*). Furthermore, dopamine-induced expression of *Bdnf* in astrocytes is controlled by an unknown regulatory region within the *Bdnf* gene locus (*Koppel et al., 2018*). Here, we identify a novel enhancer region in the *Bdnf* gene located downstream of the *Bdnf* exon III and show that this regulatory element selectively activates basal and stimulus-dependent expression of the exon I-, II-, and III-containing *Bdnf* transcripts in neurons.

## Results

### The *Bdnf* +3 kb region shows enhancer-associated characteristics in mouse and human brain tissue

To uncover novel enhancer regions regulating *Bdnf* expression in the central nervous system, we started with bioinformatic analysis of the enhancer-associated characteristics. Active enhancers are characterized by nucleosome-free DNA that is accessible to transcription factors and other DNA binding proteins. Chromatin at active enhancer regions typically has distinct histone modifications – H3K4me1, a hallmark of enhancer regions, and H3K27ac, usually associated with active regulatory regions. Active enhancers also bind RNA polymerase II and are bidirectionally transcribed from the regions marked by enhancer-associated histone modifications giving rise to non-coding enhancer RNAs (eRNAs) (*Nord and West, 2020*). Based on the mouse brain tissue chromatin immunoprecipitation sequencing (ChIP-seq) data from the ENCODE project and transcription start site (TSS) data from the FANTOM5 project, a region ~3 kb downstream of *Bdnf* exon I TSS has prominent enhancer-associated features (*Figure 1A*). First, the +3 kb region is hypersensitive to DNaseI, indicative of an open chromatin structure. Second, ChIP-seq data shows that this region is enriched for H3K4me1, H3K4me3, and H3K27ac modifications. Third, the +3 kb region interacts with RNA polymerase II, with a strong evidence for bidirectional transcription according to the FANTOM5 CAGE database. Finally, the region is conserved between mammals, pointing at its possible functional importance.

We next used H3K27ac ChIP-seq data from *Nord et al., 2013* to determine the activity of the potential enhancer region in different tissues throughout the mouse development. We found that the +3 kb region shows H3K27ac mark in the mouse forebrain, with the highest signal from embryonic day 14 to postnatal day 7, but not in the heart or liver (*Figure 1—figure supplement 1*). This suggests that the +3 kb enhancer region might be active mainly in neural tissues in late prenatal and early postnatal life.

To investigate whether the +3 kb region might also function as an enhancer in rat, we analyzed published ATAC-seq and RNA-seq data for rat cortical and hippocampal neurons (*Carullo et al., 2020*). The ATAC-seq analysis was consistent with open chromatin structure of the +3 kb region, and the RNA-seq revealed possible eRNA transcription from the antisense strand (*Figure 1—figure supplement 2*). Unfortunately, the expression of the sense-strand eRNA could not be assessed since the *Bdnf* pre-mRNA is transcribed in the same direction, masking the sense-strand eRNA signal.

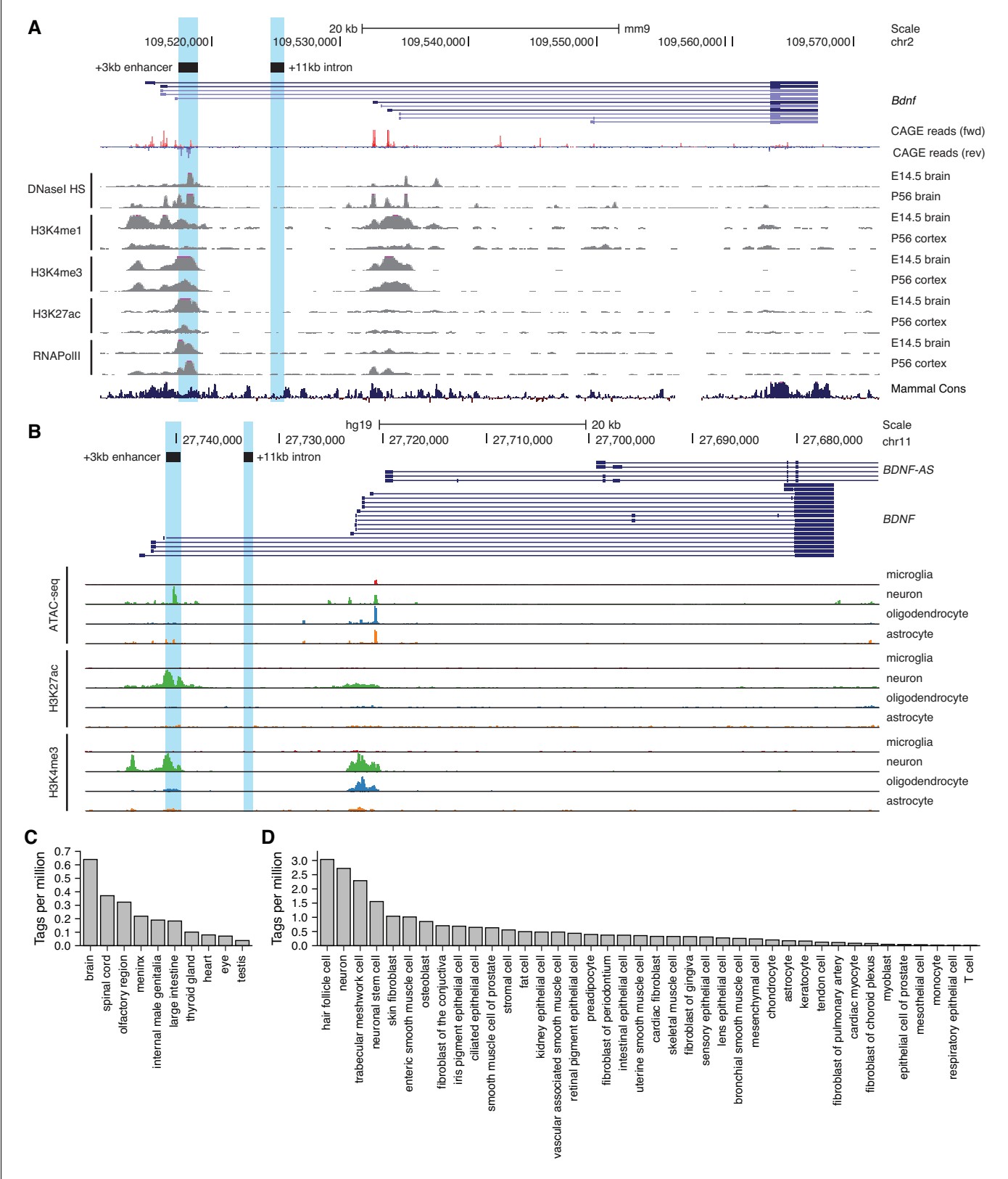

**Figure 1.** Region downstream of *Bdnf* exon III shows enhancer-associated characteristics in mouse and human neural tissues. UCSC Genome browser was used to visualize (**A**) DNaseI hypersensitivity sites and ChIP-seq data from the ENCODE project in mouse brain tissue, CAGE data of transcription start sites from the FANTOM5 project (all tissues and cell types), and (**B**) open chromatin (ATAC-seq) and ChIP-seq in different human brain cell types

*Figure 1 continued on next page*

*Figure 1 continued*

by *Nott et al., 2019*. E indicates embryonic day, P postnatal day. Signal clipping outside the visualization range is indicated with purple color. The +3 kb region, a potential enhancer of the *Bdnf* gene, and +11 kb intronic region, a negative control region used in the present study, were converted from rat genome to mouse or human genome using UCSC Liftover tool and are shown as light blue. The names of the regions represent the distance of the respective region from rat *Bdnf* exon I transcription start site. (C, D) +3 kb enhancer region (chr11:27693843–27694020, hg19 genome build) enhancer RNA (eRNA) expression levels based on CAGE sequencing data from the FANTOM5 project obtained from the Slidebase tool (*Ienasescu et al., 2016*, http://slidebase.binf.ku.dk). eRNA expression levels were grouped by different tissue types (C) or cell types (D). Only tissue and cell types with non-zero eRNA expression are shown.

The online version of this article includes the following figure supplement(s) for figure 1:

**Figure supplement 1.** The +3 kb enhancer region shows brain-specific H3K27ac histone modification.

**Figure supplement 2.** The +3 kb enhancer region shows open chromatin structure and enhancer RNA (eRNA) expression in rat cultured neurons.

Together, open chromatin and possible eRNA expression suggest that the +3 kb region could be an enhancer in rat neurons.

To further investigate which cell types the +3 kb region could be active in human in vivo, we used data from a recently published human brain cell-type-specific ATAC-seq and ChIP-seq experiments (*Nott et al., 2019*) (*Figure 1B*). We found that the +3 kb region shows remarkable neuron specificity, as evident from open chromatin identified using ATAC-seq, and H3K27ac histone mark, which are missing in microglia, oligodendrocytes, and astrocytes. To further elucidate which human tissues and cell types the +3 kb region could be active in, we used the Slidebase tool (*Ienasescu et al., 2016*) that gathers data of TSSs from FANTOM5 project and summarizes eRNA transcription levels based on various tissue and cell types. We found that in humans the +3 kb region shows the strongest eRNA expression in the brain, spinal cord, and olfactory region (*Figure 1C*). When grouped by cell type, the strongest expression of +3 kb eRNAs is in hair follicle cells, neurons, and trabecular meshwork cells (*Figure 1D*).

Collectively, this data suggests that the +3 kb region is an evolutionarily conserved nervous system-specific enhancer that is active mostly in neural tissues and predominantly in neurons but not in other major brain cell types.

## The +3 kb enhancer region shows bidirectional transcription in luciferase reporter assay in rat cultured cortical neurons and astrocytes

Based on the enhancer-associated characteristics of the +3 kb enhancer region, we hypothesized that the region could function as an enhancer region for *Bdnf* gene in neural cells. It is well known that enhancer regions can initiate RNA polymerase II-dependent bidirectional transcription of eRNAs (*Kim and Shiekhattar, 2015*; *Nord and West, 2020*; *Sartorelli and Lauberth, 2020*) and that the expression of these eRNAs is correlated with the expression of nearby genes, indicating that the transcription from an enhancer region is a proxy to enhancer's activity (*Kim et al., 2010*). Therefore, we first investigated whether the +3 kb enhancer region shows bidirectional transcription in rat cultured cortical neurons and astrocytes, the two major cell types in the brain, in a heterologous context using reporter assays. We cloned an ~1.4 kb fragment of the +3 kb enhancer and a similarly sized control (+11 kb) intronic sequence lacking enhancer-associated characteristics in either forward or reverse orientation upstream of the firefly luciferase gene (*Figure 2A*) and performed luciferase reporter assays.

In rat cortical neurons, the +3 kb enhancer region showed very strong transcriptional activity (~300-fold higher compared to the promoterless luciferase reporter vector) that was independent of the orientation of the +3 kb region (*Figure 2B*). As expected, the +11 kb negative control reporter showed very low luciferase activity in cortical neurons. To determine whether the enhancer region is responsive to different stimuli in neurons and could be involved in stimulus-dependent regulation of the *Bdnf* gene, we used two treatments shown to induce *Bdnf* gene expression – KCl treatment to chronically depolarize the cells and mimic neuronal activity (*Ghosh et al., 1994*; *Pruunsild et al., 2011*), and BDNF treatment to activate TrkB signaling and mimic BDNF autoregulation (*Esvald et al., 2020*; *Tuvikene et al., 2016*; *Yasuda et al., 2007*). Our results indicate that the activity of the +3 kb region is upregulated ~2–3-fold in response to both stimuli, suggesting that the region could be a stimulus-dependent enhancer in neurons.

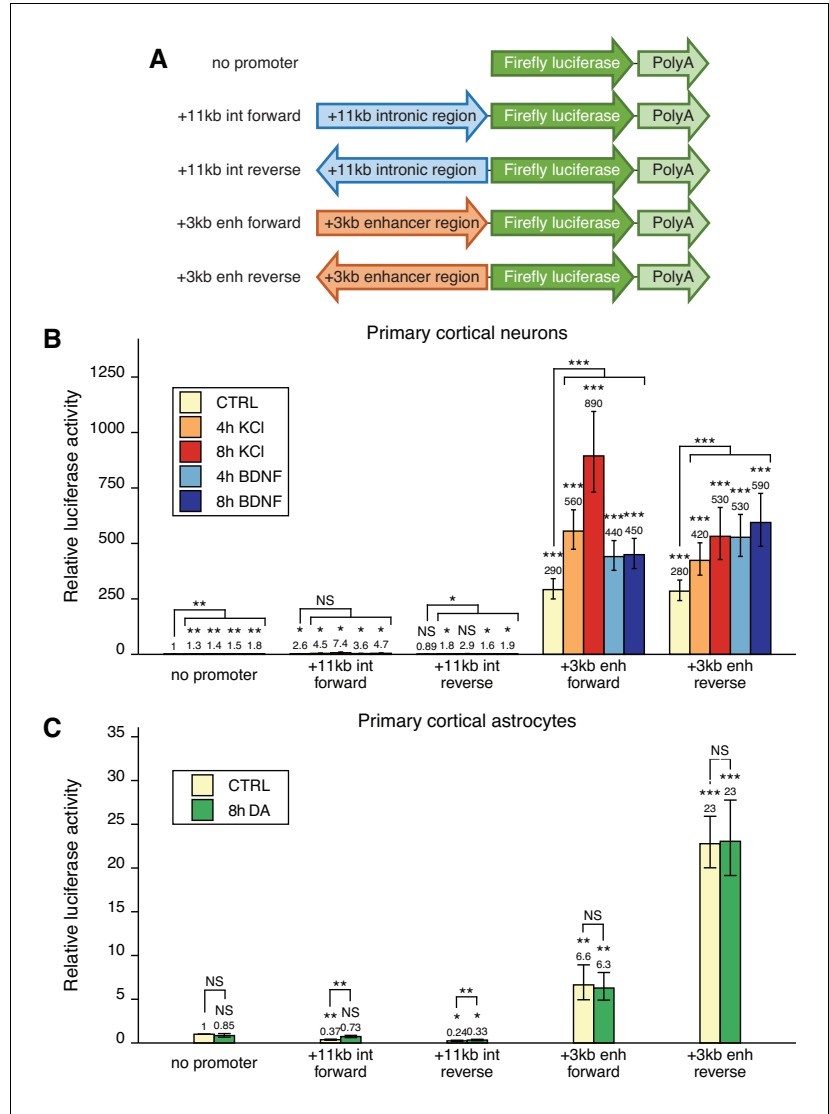

**Figure 2.** The +3 kb enhancer region shows bidirectional transcription in luciferase reporter assay in rat cultured cortical neurons and astrocytes. (**A**) Reporter constructs used in the luciferase reporter assay where the +3 kb enhancer region and the +11 kb control region were cloned in either forward or reverse orientation (respective to the rat *Bdnf* gene) in front of the luciferase expression cassette. (**B, C**) Rat cortical neurons (**B**) or astrocytes (**C**) were transfected with the indicated reporter constructs at 6 and 13 DIV, respectively. Two days post-transfection, neurons were left untreated (CTRL) or treated with 25 mM KCl (with 5 µM D-APV) or 50 ng/ml brain-derived neurotrophic factor (BDNF) for the indicated time (**B**); astrocytes were treated with 150 µM dopamine (DA) or respective volume of vehicle (CTRL) for the indicated time (**C**), after which luciferase activity was measured. Luciferase activity in cells transfected with a vector containing no promoter and treated with vehicle or left untreated was set as 1. The average luciferase activity of independent experiments is depicted above the columns. Error bars indicate SEM (n = 7 [**B**, +3 kb enhancer constructs and no promoter construct], n = 3 [**B**, intron constructs], and n = 4 [**C**] independent experiments). Asterisks above the columns indicate statistical significance relative to luciferase activity in untreated cells transfected with the reporter vector containing no promoter, or between indicated groups. NS: not significant. *p<0.05, **p<0.01, ***p<0.001 (paired two-tailed t-test).

In rat cultured cortical astrocytes, the +3 kb enhancer construct showed modest transcriptional activity (depending on the orientation ~6–23-fold higher compared to the promoterless vector control, *Figure 2C*). We have previously shown that in cultured cortical astrocytes *Bdnf* is induced in response to dopamine treatment, and the induction is regulated by an unknown enhancer region within *Bdnf* gene locus (*Koppel et al., 2018*). However, dopamine treatment had no significant

effect on the activity of the +3 kb construct in cultured cortical astrocytes, suggesting that this region is not a dopamine-activated enhancer in astrocytes.

## The +3 kb enhancer region potentiates the activity of *Bdnf* promoters I and IV in luciferase reporter assay in rat cultured cortical neurons

To find out whether the newly identified +3 kb enhancer could control the activity of *Bdnf* promoters in a heterologous context, we cloned the +3 kb region or the +11 kb negative control sequence in forward or reverse orientation (respective to the rat *Bdnf* gene) downstream of *Bdnf* promoter-driven luciferase expression cassette (*Figure 3A*).

First, we transfected rat cultured cortical neurons with constructs containing *Bdnf* promoter I or IV as these promoters are the most widely studied in neurons (*West et al., 2014*). We treated neurons with either KCl or BDNF and used luciferase reporter assay to measure the activity of the *Bdnf* promoter region. For *Bdnf* promoter I (*Figure 3B*), the addition of the +3 kb enhancer region slightly increased the basal activity of the promoter region (~1.5–2-fold). The +3 kb enhancer region also potentiated the KCl- and BDNF-induced activity of this promoter approximately threefold in an orientation-independent manner. Similar effects were observed for *Bdnf* promoter IV (*Figure 3C*), where the addition of the +3 kb enhancer region potentiated the basal activity of the promoter approximately threefold and KCl- and BDNF-induced activity levels approximately twofold. The +11 kb intronic region failed to potentiate the activity of *Bdnf* promoters I and IV.

Cortical astrocytes preferentially express *Bdnf* transcripts containing 5' exons IV and VI (*Koppel et al., 2018*). Notably, the +3 kb enhancer failed to increase the activity of *Bdnf* promoters IV and VI in unstimulated astrocytes and had little effect on the response of these promoters to dopamine (*Figure 3D, E*).

Overall, we found that in the heterologous context the +3 kb enhancer region could potentiate transcription from *Bdnf* promoters in cultured cortical neurons but not in cortical astrocytes. These results imply that the +3 kb enhancer could be important for *Bdnf* gene expression in neurons but not in astrocytes.

## The +3 kb enhancer region is a positive regulator of the first cluster of *Bdnf* transcripts in rat cortical neurons but is in an inactive state in rat cortical astrocytes

To investigate the functionality of the +3 kb region in its endogenous context, we used CRISPR interference (CRISPRi) and activator (CRISPRa) systems. Our system comprised catalytically inactive Cas9 (dCas9) fused with Krüppel-associated box domain (dCas9-KRAB, CRISPRi), or 8 copies of VP16 activator domain (VP64-dCas9-VP64, CRISPRa) to repress or activate the target region, respectively. dCas9 without effector domains was used to control for potential steric effects (*Qi et al., 2013*) on *Bdnf* transcription when targeting CRISPR complex inside the *Bdnf* gene. To direct the dCas9 and its variants to the desired location, we used four different gRNAs per region targeting either the +3 kb enhancer region or the +11 kb intronic control region, with all gRNAs targeting the template strand to minimize the potential inhibitory effect of dCas9 binding on transcription elongation (as suggested by *Qi et al., 2013*). The +11 kb intronic control was used to rule out the possibility of CRISPRi- and CRISPRa-induced passive spreading of chromatin modifications within the *Bdnf* gene locus. As a negative control, we used a gRNA not corresponding to any sequence in the rat genome.

We first examined the functionality of the +3 kb enhancer region in cultured cortical neurons. Targeting the +3 kb enhancer or +11 kb intronic region with dCas9 without an effector domain had no major effect on the expression of any of the *Bdnf* transcripts, indicating that targeting CRISPR complex to an intragenic region in *Bdnf* gene does not itself notably affect *Bdnf* gene expression (*Figure 4*, left panel). Repressing the +3 kb enhancer region using CRISPRi decreased the basal expression levels of *Bdnf* exon I-, IIc-, and III-containing transcripts by 2.2-, 11-, and 2.4-fold, respectively, although these effects were not statistically significant (*Figure 4A–C*, middle panel). In contrast, no notable effect was seen for basal levels of *Bdnf* exon IV-, VI-, and IXa-containing transcripts (*Figure 4D–F*, middle panel). Repressing the +3 kb enhancer region also decreased the KCl and BDNF-induced levels of transcripts starting from the first three 5' exons ~4–7-fold, but not of other

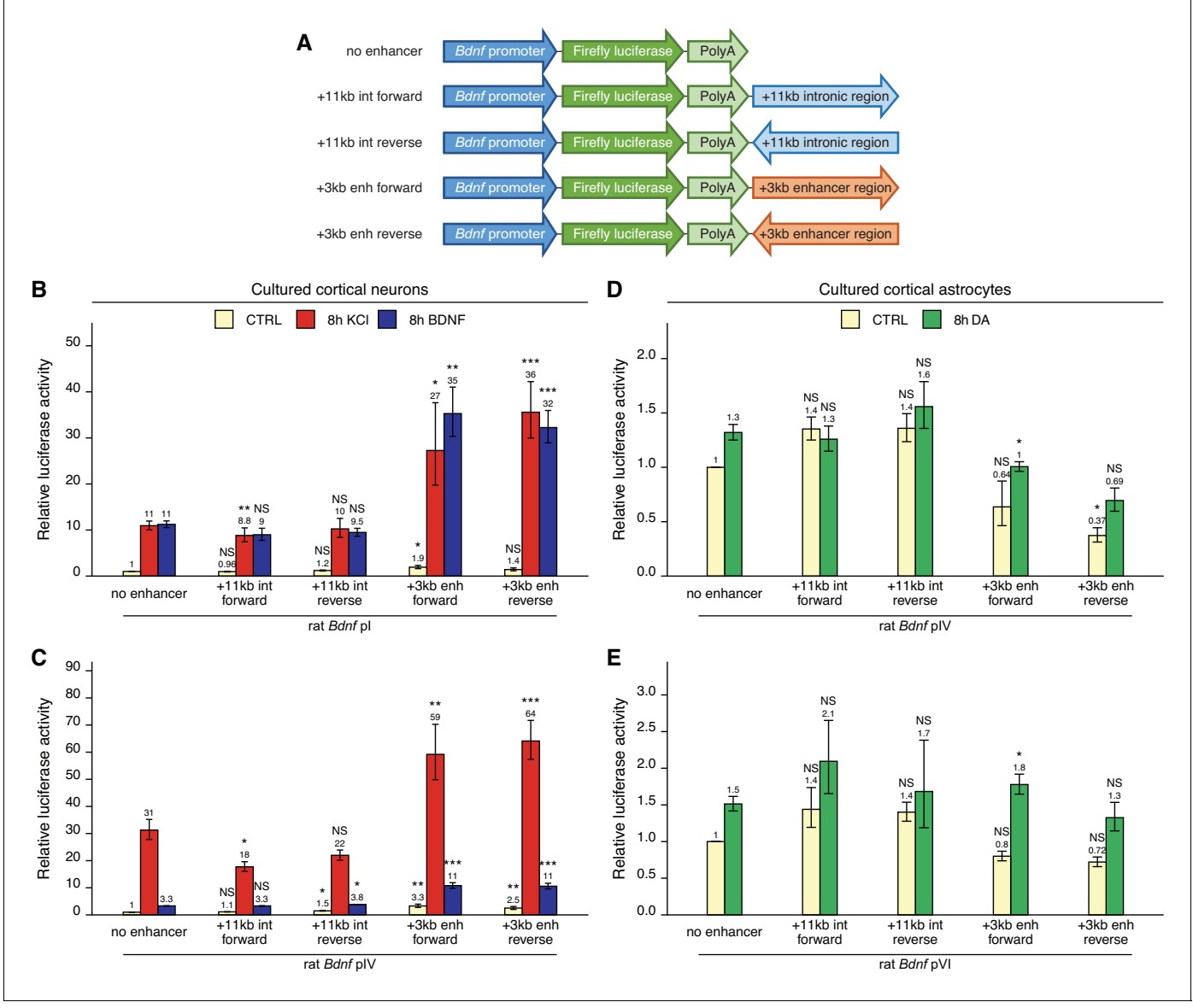

**Figure 3.** The +3 kb enhancer region potentiates the activity of *Bdnf* promoters in luciferase reporter assay in rat cortical neurons but not in astrocytes. (A) A diagram of the luciferase reporter constructs used in this experiment, with a *Bdnf* promoter in front of the firefly luciferase coding sequence and the +3 kb enhancer or +11 kb intronic region in either forward or reverse orientation (respective to the rat *Bdnf* gene) downstream of the luciferase expression cassette. Rat cortical neurons (B, C) or astrocytes (D, E) were transfected with the indicated reporter constructs at 6 and 13 DIV, respectively. Two days post transfection, neurons were left untreated (CTRL) or treated with 25 mM KCl (with 5 µM D-APV) or 50 ng/ml brain-derived neurotrophic factor (BDNF) for 8 hr (B, C); astrocytes were treated with 150 µM dopamine (DA) or respective volume of vehicle (CTRL) for 8 hr (D, E), followed by luciferase activity assay. Luciferase activity is depicted relative to the luciferase activity in untreated or vehicle-treated (CTRL) cells transfected with respective *Bdnf* promoter construct without an enhancer region. The average luciferase activity of independent experiments is shown above the columns. Error bars represent SEM (n = 6 [B, +3 kb enhancer-containing constructs and no enhancer construct], n = 3 [B, + 11 kb intron constructs], n = 4 [C], and n = 3 [D–E] independent experiments). Statistical significance was calculated compared to the activity of the respective *Bdnf* promoter regions without the enhancer region after the respective treatment. NS: not significant. *p<0.05, **p<0.01, ***p<0.001 (paired two-tailed t-test).

*Bdnf* transcripts (*Figure 4A–F*, middle panel). These effects correlated with subtle changes in total *Bdnf* expression levels (*Figure 4G*, middle panel). Targeting CRISPRi to the +11 kb intronic region had no notable effect on any of the *Bdnf* transcripts (*Figure 4A–F*, middle panel).

Activating the +3 kb enhancer region with CRISPRa in cultured cortical neurons increased the expression levels of *Bdnf* transcripts of the first cluster (*Figure 4A–C*, right panel) both in unstimulated neurons (~3–5-fold) and after KCl or BDNF treatment (~2-fold). Slight effect of the activation

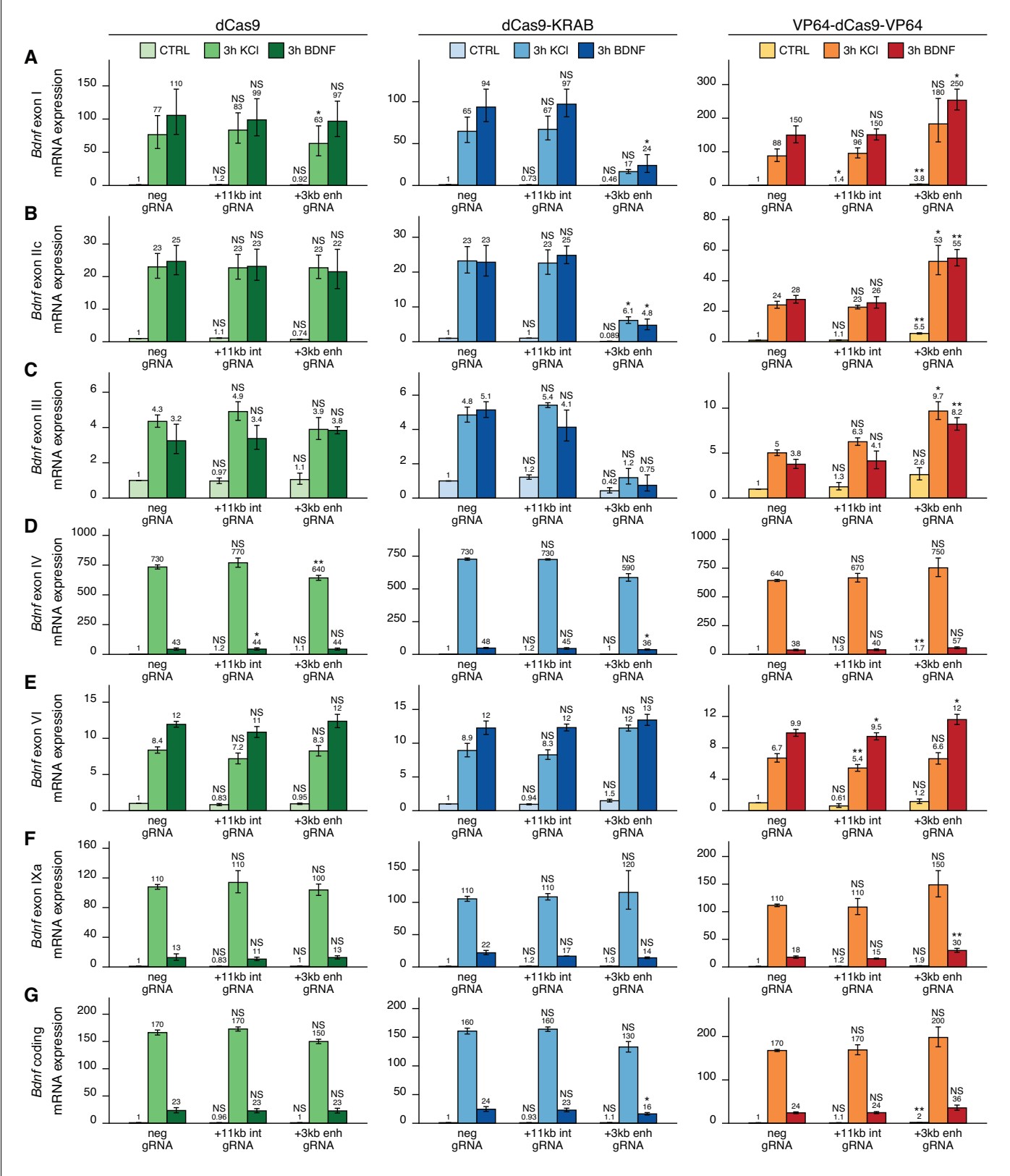

**Figure 4.** The +3 kb enhancer is a positive regulator of *Bdnf* exon I-, IIc-, and III-containing transcripts in rat cortical neurons. Rat cultured cortical neurons were transduced at 0 DIV with lentiviral particles encoding either catalytically inactive Cas9 (dCas9, left panel, green), dCas9 fused with Krüppel-associated box domain (dCas9-KRAB, middle panel, blue), or 8 copies of VP16 activator domain (VP64-dCas9-VP64, right panel, orange),

*Figure 4 continued on next page*

*Figure 4 continued*

together with lentiviruses encoding either guide RNA that has no corresponding target sequence in the rat genome (neg gRNA), a mixture of four gRNAs directed to the putative +3 kb *Bdnf* enhancer (+3 kb enh gRNA), or a mixture of four gRNAs directed to +11 kb intronic region (+11 kb int gRNA). Transduced neurons were left untreated (CTRL) or treated with 50 ng/ml brain-derived neurotrophic factor (BDNF) or 25 mM KCl (with 5 µM D-APV) for 3 hr at 8 DIV. Expression levels of different Bdnf transcripts (**A-F**) or total *Bdnf* (**G**) were measured with RT-qPCR. mRNA expression levels are depicted relative to the expression of the respective transcript in untreated (CTRL) neurons transduced with negative guide RNA within each set (dCas9, dCas9-KRAB, or VP64-dCas9-VP64). The average mRNA expression of independent experiments is depicted above the columns. Error bars represent SEM (n = 3 independent experiments). Statistical significance was calculated between the respective mRNA expression levels in respectively treated neurons transduced with neg gRNA within each set (dCas9, Cas9-KRAB, or VP64-dCas9-VP64). NS: not significant. *$p<0.05$, **$p<0.01$, ***$p<0.001$ (paired two-tailed t-test).

The online version of this article includes the following figure supplement(s) for figure 4:

**Figure supplement 1.** The +3 kb enhancer shows stimulus-dependent enhancer RNA (eRNA) transcription in neurons.

**Figure supplement 2.** The +3 kb enhancer shows membrane depolarization-induced enhancer RNA (eRNA) transcription in rat cultured cortical and hippocampal neurons.

was also seen for *Bdnf* exon IV- and IXa-containing transcripts (*Figure 4D, F*, right panel). Total *Bdnf* basal levels also increased ~2-fold with CRISPR activation of the +3 kb enhancer region, whereas a slightly weaker effect was seen in stimulated neurons. As with CRISPRi, targeting CRISPRa to the +11 kb intronic region had no dramatic effect on the expression of any of the *Bdnf* transcripts, despite some minor changes reaching statistical significance (*Figure 4A–F*, right panel).

Next, we carried out CRISPRi and CRISPRa experiments in cultured cortical astrocytes. Targeting dCas9 to the *Bdnf* locus did not affect the expression of most *Bdnf* transcripts (*Figure 5A and C–F*, left panel), although a slight decrease could be seen for exon IIc-containing transcripts in cells treated with dopamine (*Figure 5B*). Targeting CRISPRi to the +3 kb enhancer region in astrocytes did not decrease the basal expression levels of any *Bdnf* transcript (*Figure 5*, middle panel). In cells treated with dopamine, repressing +3 kb enhancer region completely abolished the induction of exon IIc-containing transcripts (*Figure 5B*, middle panel), but did not affect the expression of any other transcripts. In contrast, activating the +3 kb region with CRISPRa greatly increased both basal and dopamine-induced levels of all measured *Bdnf* transcripts (*Figure 5A–F*, right panel). Targeting CRISPRi or CRISPRa to the +11 kb intronic control region did not have a noteworthy effect on any of the measured *Bdnf* transcripts.

As bioinformatic analysis showed bidirectional transcription from the +3 kb enhancer (*Figure 1*) and our luciferase reporter assays also indicated this (*Figure 2*), we next decided to directly measure eRNAs from the +3 kb enhancer region in our cultured cortical neurons and astrocytes. Since the sense eRNA is transcribed in the same direction as *Bdnf* pre-mRNA, we could only reliably measure eRNAs from the antisense orientation from the +3 kb enhancer region using antisense eRNA-specific cDNA priming followed by qPCR (*Figure 4—figure supplement 1*). We found that the +3 kb enhancer antisense eRNA was expressed in cultured neurons and the expression level of the eRNA was induced ~3.5- and ~6-fold upon BDNF and KCl treatment, respectively. Furthermore, repressing the +3 kb enhancer region using CRISPRi decreased the expression of the eRNA ~3-fold. However, activating the +3 kb enhancer region using CRISPRa did not change the expression level of the eRNA. We also analyzed RNA-seq data from *Carullo et al., 2020* and confirmed the membrane depolarization-dependent induction of +3 kb antisense eRNA expression in rat cultured cortical and hippocampal neurons (*Figure 1—figure supplement 2*, *Figure 4—figure supplement 2*). When comparing +3 kb enhancer eRNA expression levels in neurons and astrocytes, the astrocytes showed ~6-fold lower eRNA transcription from the +3 kb enhancer region than neurons (*Figure 5—figure supplement 1*), also indicating that the +3 kb region is in a more active state in our cultured neurons than in astrocytes.

The main findings of the CRISPRi and CRISPRa experiments are summarized in *Figure 6*. Taken together, our results suggest that the +3 kb enhancer region is an active enhancer of the *Bdnf* gene in rat cultured cortical neurons and regulates the basal and stimulus-induced expression of *Bdnf* transcripts of the first cluster of exons (exons I, II, and III). In contrast, the +3 kb region is mostly inactive in rat cultured cortical astrocytes.

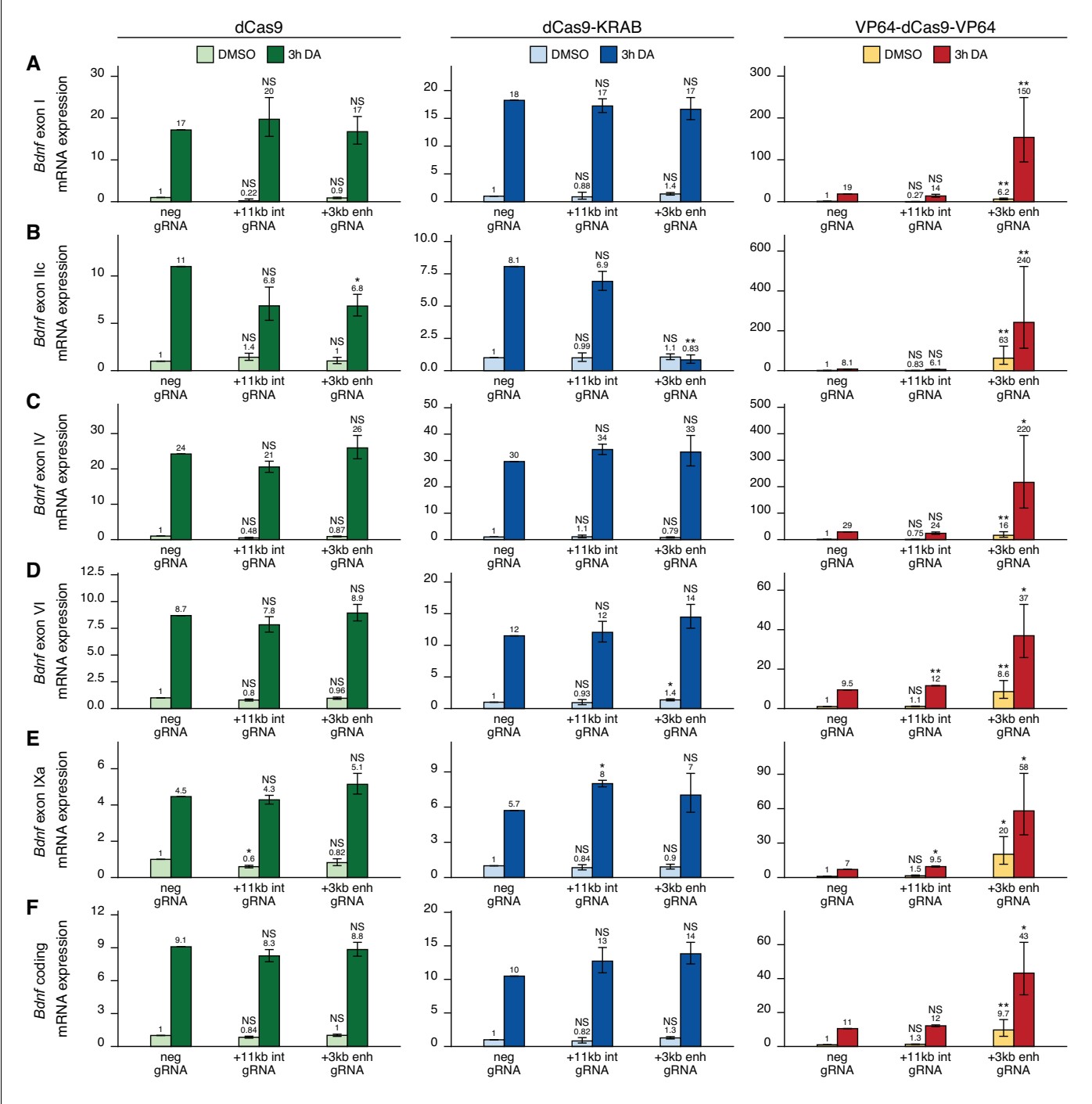

**Figure 5.** The +3 kb enhancer region is mainly inactive in rat cortical astrocytes. Rat cultured cortical astrocytes were transduced at 7 DIV with lentiviral particles encoding either catalytically inactive Cas9 (dCas9, left panel, green), dCas9 fused with Krüppel-associated box domain (dCas9-KRAB, middle panel, blue), or 8 copies of VP16 activator domain (VP64-dCas9-VP64, right panel, orange), together with lentiviruses encoding either guide RNA that has no corresponding target sequence in the rat genome (neg gRNA), a mixture of four gRNAs directed to the putative +3 kb *Bdnf* enhancer (+3 kb enh gRNA), or a mixture of four guide RNAs directed to the + 11 kb intronic region (+11 kb int gRNA). Transduced astrocytes were treated with vehicle (CTRL) or with 150 µM dopamine (DA) for 3 hr at 15 DIV. Expression levels of different *Bdnf* transcripts (A-E) or total *Bdnf* (F) were measured with RT-qPCR. The levels of *Bdnf* exon III-containing transcripts were too low to measure reliably. mRNA expression levels are depicted relative to the expression of the respective transcript in astrocytes treated with vehicle (CTRL) transduced with negative guide RNA within each set (dCas9, dCas9-KRAB, or VP64-dCas9-VP64). The average mRNA expression of independent experiments is depicted above the columns. Error bars represent SEM (n = 3 independent experiments). Statistical significance was calculated between respective mRNA expression levels in respectively treated astrocytes

*Figure 5 continued on next page*

*Figure 5 continued*

transduced with neg gRNA within each set (dCas9, Cas9-KRAB, or VP64-dCas9-VP64). NS: not significant. *p<0.05, **p<0.01, ***p<0.001 (paired two-tailed t-test).

The online version of this article includes the following figure supplement(s) for figure 5:

**Figure supplement 1.** The +3 kb enhancer shows lower enhancer RNA (eRNA) expression in astrocytes than in neurons.

### Deletion of the +3 kb enhancer region in mouse embryonic stem cell-derived neurons decreases the expression of *Bdnf* transcripts starting from the first cluster of exons

To test the regulatory function of the +3 kb enhancer directly and address biological significance of its interspecies conservation, we used CRISPR/Cas9 system to delete the conserved core sequence of the enhancer region in mouse embryonic stem cells (mESC) engineered to express the pro-neural transcription factor Neurogenin2 from a doxycycline-inducible promoter. We selected single-cell clones containing the desired deletion (*Figure 7—figure supplement 1*), differentiated them into neurons (*Figure 7—figure supplement 2*, *Ho et al., 2016*; *Thoma et al., 2012*; *Zhang et al., 2013*) by the addition of doxycycline, treated the stem cell-derived neurons with BDNF or KCl, and measured the expression of different *Bdnf* transcripts using RT-qPCR (*Figure 7*).

The deletion of the +3 kb enhancer region strongly decreased both the basal and stimulus-dependent expression levels of *Bdnf* exon I-, IIc-, and III-containing transcripts (*Figure 7A–C*). Notably, the effect was more prominent in clones containing homozygous deletion compared to heterozygous clones. We also noted a slight, albeit not statistically significant decrease in the expression of

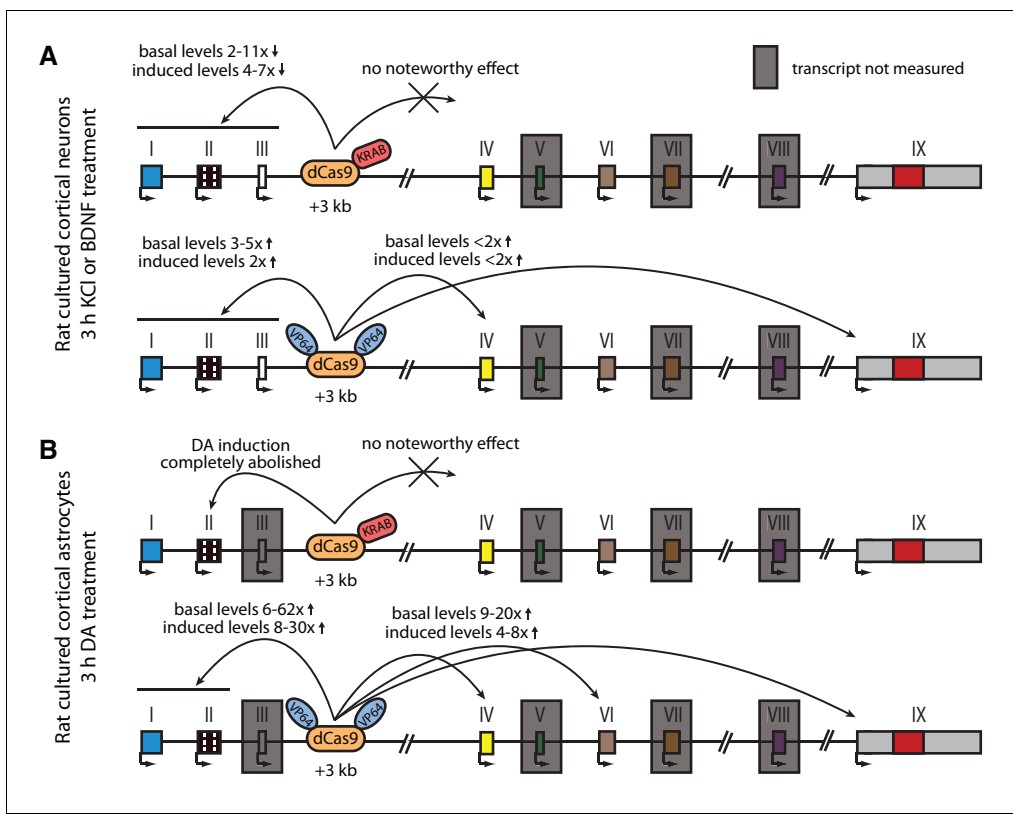

**Figure 6.** Summary of the CRISPRi and CRISPRa experiments with *Bdnf* +3 kb enhancer region in rat cultured cortical neurons and astrocytes. Graphical representation of the main results shown in *Figures 4* (A) and  *5* (B). Different *Bdnf* exons are shown with boxes, and red box in exon IX indicates *Bdnf* coding region. *Bdnf* transcripts that were not measured or that had too low levels to measure reliably are indicated with a gray box around the respective 5' exon.

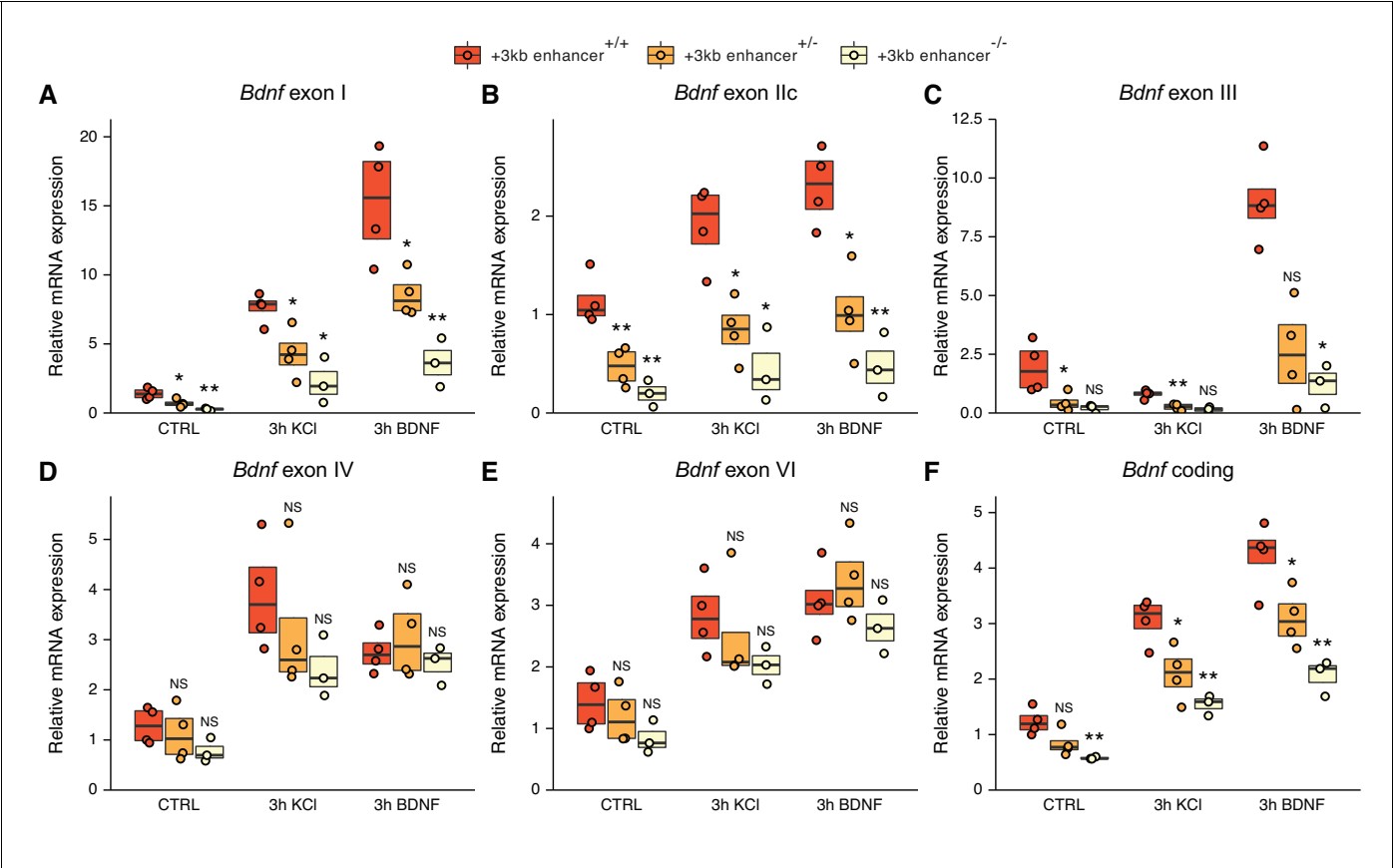

**Figure 7.** The deletion of the +3 kb enhancer region decreases the expression of *Bdnf* exon I-, IIc-, and III-containing transcripts in mouse embryonic stem cell (mESC)-derived neurons. CRISPR/Cas9 system was used to generate mESC cell lines with ~300–500 bp deletions of the conserved core region of the +3 kb *Bdnf* enhancer. The obtained clonal cell lines containing intact +3 kb enhancer region (+/+), heterozygous deletion (+/–), or homozygous deletion (–/–) of the +3 kb enhancer region were differentiated into neurons using overexpression of Neurogenin2. After 12 days of differentiation, the cells were treated with vehicle (CTRL), 50 ng/ml brain-derived neurotrophic factor (BDNF) or 25 mM KCl together with 25 µM D-APV for 3 hr. The expression levels of different *Bdnf* transcripts (A-E) or total *Bdnf* (F) were measured using RT-qPCR. The levels of respective *Bdnf* transcripts measured in the parental cell line (also included as a data point in the +/+ group) were set as 1. All data points (obtained from independent cell clones and parental cell line) are depicted with circles. Box plot shows 25% and 75% quartiles, and the horizontal line shows the median value. N = 3–4 independent cell clones for each group. Statistical significance was calculated compared to the expression level of the respective transcript in the +/+ genotype group at respective treatment. NS: not significant. *p<0.05, **p<0.01, ***p<0.001 (equal variance unpaired t-test).

The online version of this article includes the following figure supplement(s) for figure 7:

**Figure supplement 1.** Verification of the deletion of the +3 kb enhancer region in mouse embryonic stem cell (mESC) clones.

**Figure supplement 2.** Characterization of mouse embryonic stem cell (mESC) differentiation into neurons.

*Bdnf* exon IV- and VI-containing transcripts in cells, where the +3 kb enhancer region was deleted (*Figure 7D–E*). This could be attributed to impaired BDNF autoregulatory loop caused by the deficiency of transcripts from the first cluster of exons. It is also possible that the +3 kb enhancer participates in the regulation of the transcripts from the second cluster of exons, but the effect is only very subtle, which could explain why it was not detected in our CRISPRi and CRISPRa experiments in cultured neurons. The levels of *Bdnf* exon IXa-containing transcripts were too low to measure reliably (data not shown). The deletion of the +3 kb enhancer region decreased the total levels of *Bdnf* similarly to the first cluster of *Bdnf* transcripts (*Figure 7F*). These results confirm the essential role of the +3 kb enhancer region in regulating the expression of *Bdnf* exon I-, IIc-, and III-containing transcripts in rodent neurons.

## The activity of the +3 kb enhancer region is regulated by CREB, AP-1 family and E-box-binding transcription factors

To investigate the molecular mechanisms that control the activity of the +3 kb enhancer region, we used in vitro DNA pulldown assay with 8-day-old rat cortical nuclear lysates coupled with mass spectrometric analysis to determine the transcription factors that bind to the +3 kb enhancer region (*Figure 8A*). Collectively, we determined 21 transcription factors that showed specific in vitro binding to the +3 kb enhancer region compared to the +11 kb intronic region in two independent experiments (*Figure 8B*). Of note, we found numerous E-box-binding proteins, including USFs, TCF4 and the pro-neural transcription factors NeuroD2 and NeuroD6, possibly providing the neuron-specific activity of the +3 kb enhancer region. We also detected binding of JunD, a member of the AP-1 transcription factor family.

Next, we used various ChIP-seq experiments in different human cell lines from the ENCODE project and determined numerous transcription factors that bind to the +3 kb enhancer region, including CREB, CEBPB, EGR1, and JunD (*Figure 8—figure supplement 1*). We further used publicly available ChIP-seq data (see 'Materials and methods' section for references) to visualize the binding of different transcription factors to the +3 kb enhancer region in mouse neural cells and tissues (*Figure 8C*). This data shows neuronal activity-dependent binding of NPAS4, c-Fos, and coactivator CBP to the enhancer region. In agreement with our in vitro pulldown results, ChIP-seq analysis also revealed binding of EGR1, NeuroD2, and TCF4 to the +3 kb enhancer region in the endogenous chromatin context.

Considering the CREB binding in ENCODE data (*Figure 8—figure supplement 1*) and CBP binding in cultured cortical neurons (*Figure 8C*), we first investigated whether CREB binds *Bdnf* +3 kb enhancer region in our rat cortical neurons. We performed ChIP-qPCR and determined that in cultured cortical neurons CREB binds to the +3 kb enhancer region, whereas we found no significant CREB binding to the +21 kb negative control region, located directly downstream of *Bdnf* exon VII (*Figure 8D*). Of note, the binding of CREB to the +3 kb enhancer region was ~2.6 times stronger than binding to *Bdnf* promoter IV, which contains the well-described CRE element (*Hong et al., 2008*; *Tao et al., 1998*). Next, we focused on the various E-box-binding proteins as many E-box-binding proteins are pro-neural and could therefore confer the neural specificity of the +3 kb enhancer region. As transcription factors from the NeuroD family need dimerization partner from the class I helix–loop–helix proteins, for example, TCF4, to bind DNA (*Massari and Murre, 2000*; *Ravanpay and Olson, 2008*), we verified the binding of TCF4 to the +3 kb region in our cultured neurons using TCF4 ChIP-qPCR (*Figure 8D*).

To determine functionally important transcription factors that regulate *Bdnf* +3 kb enhancer region, we first screened a panel of dominant-negative transcription factors in luciferase reporter assay where the expression of luciferase was under control of the +3 kb enhancer region (*Figure 8E*). In agreement with the in vitro pulldown assay, ChIP-seq and ChIP-qPCR results, we found the strongest inhibitory effect using dominant negative versions of CREB (named A-CREB), ATF2 (named A-ATF2), and AP-1 family (named A-FOS). The effect of different dominant negative proteins was slightly lower when the +3 kb enhancer region was in the reverse orientation. Our data suggests the role of CREB, AP-1 family proteins, and ATF2 in regulating the neuronal activity-dependent activation of the +3 kb enhancer region, whereas we found no notable evidence of USF family transcription factors and CEBPB regulating the activity of the +3 kb enhancer region.

Finally, we elucidated the role of E-box-binding proteins in the regulation of the +3 kb enhancer region. Using luciferase reporter assays, we found that silencing TCF4 expression with TCF4 shRNA-expressing plasmid decreased the activity of the +3 kb enhancer region in both unstimulated and KCl- and BDNF-stimulated neurons. However, the effects were slightly smaller when the enhancer region was in the reverse orientation (*Figure 8F*). Based on the TCF4 and NeuroD2 ChIP-seq data (*Figure 8C*), we identified a putative E-box binding sequence in the +3 kb enhancer region (CAGATG). To determine the relevance of this E-box element, we generated +3 kb enhancer-containing reporter constructs where this E-box motif was mutated (CAGAAC). We determined that this motif participates in regulating both the basal activity and BDNF- and KCl-induced activity of the enhancer region (*Figure 8G*). Importantly, mutating the E-box decreased the ability of the +3 kb enhancer region to potentiate transcription from *Bdnf* promoter I in reporter assays (*Figure 8H*).

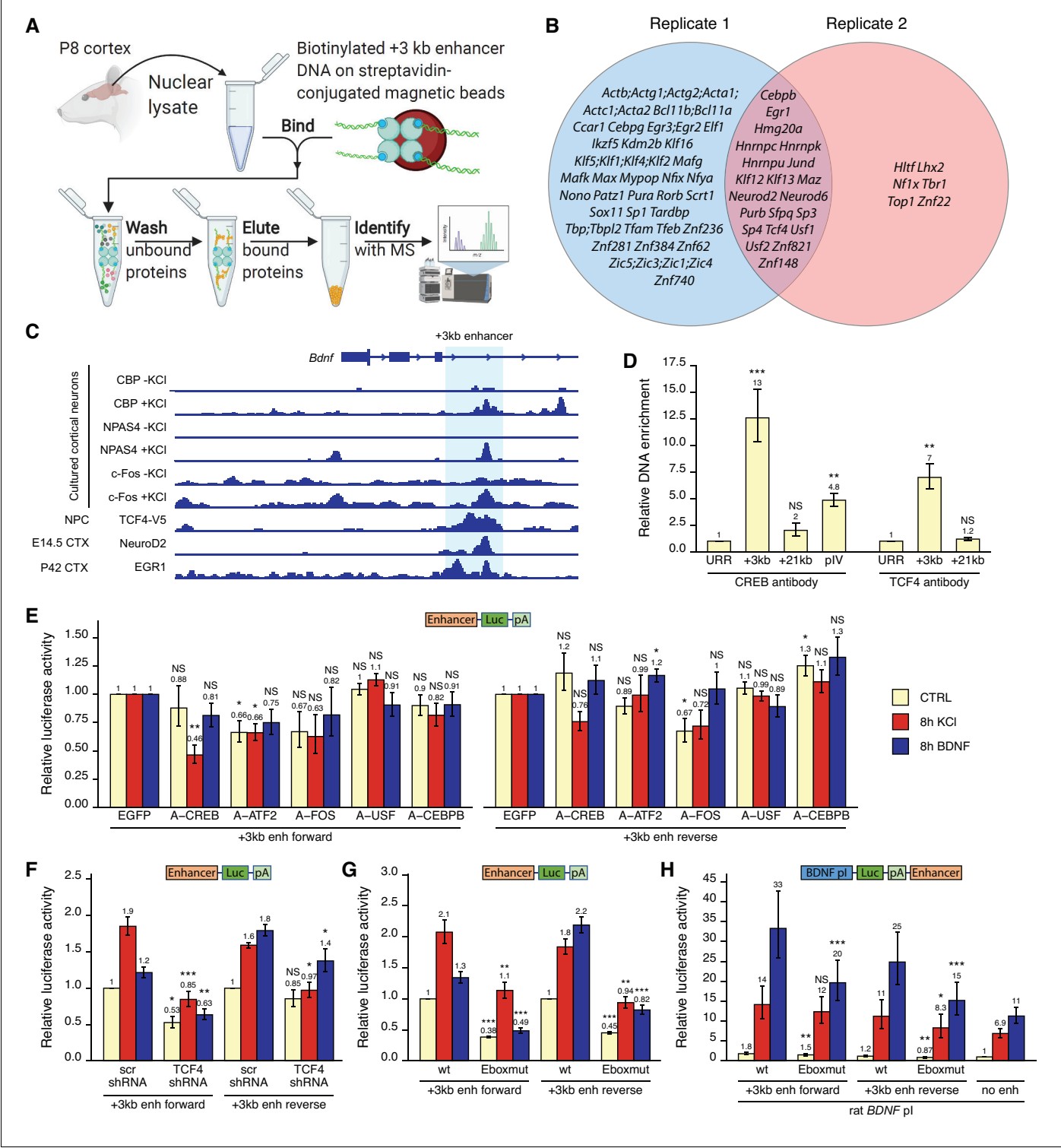

**Figure 8.** Various transcription factors, including CREB, AP-1 proteins, and E-box-binding transcription factors, regulate the activity of the +3 kb enhancer region. (**A**) Schematic overview of the in vitro DNA pulldown assay to determine transcription factors binding to the +3 kb enhancer region. The illustration was created with BioRender.com. (**B**) Gene names of the transcription factors identified in the in vitro DNA pulldown assay in two biological replicates of postnatal day 8 (P8) rat cortices. Semicolon between gene names indicates uncertainty in the peptide to protein assignment between the genes separated by the semicolons. (**C**) Previously published ChIP-seq experiments showing binding of different transcription factors to the +3 kb enhancer region. (**D**) ChIP-qPCR assay in cultured cortical neurons at 8 DIV with anti-CREB or anti-TCF4 antibody. Enrichment is shown relative to the enrichment of unrelated region (URR) with the respective antibody. The +21 kb region (downstream of the *Bdnf* exon VII) was used as a

*Figure 8 continued on next page*

*Figure 8 continued*

negative control. pIV indicates *Bdnf* promoter IV region. (**E–H**) Rat cortical neurons were transfected at 5 DIV (**F**) or 6 DIV (**E–H**) with reporter constructs where the +3 kb enhancer region was cloned in front of the luciferase coding sequence (**E–G**, see also *Figure 2A*), or with reporter constructs where the +3 kb enhancer region was cloned downstream of the *Bdnf* promoter I-controlled firefly luciferase expression cassette (**H**, see *Figure 3A*). Schematic representations of the used reporter constructs are shown above the graphs, with Luc designating luciferase coding sequence and pA polyadenylation sequence. At 8 DIV, neurons were left untreated (CTRL) or treated with 25 mM KCl (with 5 µM D-APV) or 50 ng/ml brain-derived neurotrophic factor (BDNF) for the indicated time, after which luciferase activity was measured. Luciferase activity is depicted relative to the luciferase activity in respectively treated cells transfected with enhanced green fluorescent protein (EGFP)-encoding plasmid and the respective +3 kb enhancer construct (**E**), relative to the luciferase activity in untreated cells co-transfected with control shRNA (scr) and the respective +3 kb enhancer construct (**F**), relative to the luciferase activity in untreated cells transfected with the respective wild-type (wt) +3 kb enhancer construct (**G**), or relative to the luciferase activity in untreated cells transfected with rat *Bdnf* promoter I construct containing no enhancer region (**H**). Eboxmut indicates mutation of a putative E-box element in the +3 kb enhancer region. Numbers above the columns indicate average, error bars represent SEM (n = 4 [**D**, CREB antibody], n = 3 [**D**, TCF4 antibody], n = 5–6 [**E**], n = 4–5 [**F**], n = 4 [**G**], and n = 7 [**H**] independent experiments). Statistical significance was calculated compared to the ChIP enrichment of DNA at the URR region using respective antibody (**D**), compared to the luciferase activity in respectively treated cells transfected with the respective +3 kb enhancer construct and EGFP (**E**), scr shRNA (**F**), or the respective wt +3 kb enhancer construct (**G**, **H**). NS: not significant. *p<0.05, **p<0.01, ***p<0.001 (paired two-tailed t-test).

The online version of this article includes the following figure supplement(s) for figure 8:

**Figure supplement 1.** The +3 kb enhancer region binds various transcription factors in human cell lines.

Collectively, we have identified numerous transcription factors that potentially regulate the activity of the +3 kb enhancer region and further discovered a functional E-box element in the enhancer, possibly conferring neuron-specific activity of the +3 kb enhancer region.

## Discussion

*Bdnf* promoters I, II, and III are located within a relatively compact (~2 kb) region in the genome, making it possible that their activity is controlled by a common mechanism. Spatial clustering of *Bdnf* exons seems to be conserved in vertebrates (*Keifer, 2021*), with a similar genomic organization observed in frog (*Kidane et al., 2009*), chicken (*Yu et al., 2009*), zebrafish (*Heinrich and Pagtakhan, 2004*), rodents (*Aid et al., 2007*), and human (*Pruunsild et al., 2007*). It has previously been suggested that *Bdnf* promoters I and II could be co-regulated as one functional unit (*Hara et al., 2009*; *Timmusk et al., 1999*; *West et al., 2014*). Here, we show that the promoters of *Bdnf* exons I, II, and III are co-regulated as a neuron-specific unit through a conserved enhancer region located downstream of exon III.

We have previously reported that exon I-containing *Bdnf* transcripts contain in-frame alternative translation start codon that is used more efficiently for translation initiation than the canonical start codon in exon IX (*Koppel et al., 2015*). As the *Bdnf* exon I-containing transcripts are highly inducible in response to different stimuli, they could make a substantial contribution to the overall production of BDNF protein in neurons, despite the low basal expression levels of this transcript. Remarkably, the *Bdnf* transcripts from the first cluster of exons have been shown to regulate important aspects of behavior. In female mice, *Bdnf* exon I-containing transcripts are important for proper sexual and maternal behavior (*Maynard et al., 2018*), whereas in male mice the *Bdnf* exon I- and exon II-containing transcripts regulate serotonin signaling and control aggressive behavior (*Maynard et al., 2016*). Furthermore, it has been shown that *Bdnf* exon I-containing transcripts in the hypothalamus participate in energy metabolism and thermoregulation (*You et al., 2020*). The +3 kb enhancer identified in our work might therefore be an important regulator of *Bdnf* gene expression in the formation of the neural circuits regulating both social behavior and energy metabolism. Further work will address this possibility experimentally.

The data from *Nord et al., 2013* indicates that the highest H3K27ac modification, a hallmark of active regulatory region, at the +3 kb enhancer region in development occurs a week before and a week after birth in mice – coinciding with the period of late neurogenesis, neuronal migration, synaptogenesis, and maturation of neurons (*Reemst et al., 2016*). It appears that the +3 kb enhancer region is mostly active in early life and participates in the development of the central nervous system via regulating *Bdnf* expression. However, it is also possible that the decline in H3K27ac mark in rodent brain tissue during postnatal development is due to the increased amount of non-neuronal

cells in the brain compared to neurons. Although the activity of the +3 kb enhancer seems to decrease with age, it is plausible that it remains active also in later postnatal life and upregulates *Bdnf* expression, thereby regulating synaptic plasticity in the adult organism.

Based on the induction of eRNA expression from the +3 kb enhancer region upon depolarization and BDNF-TrkB signaling in both luciferase reporter assays and in the endogenous context, and binding of various activity-dependent transcription factors to the +3 kb region, our data indicates that, in addition to conferring neuron specificity, the +3 kb enhancer region also participates in BDNF-TrkB signaling- and neuronal activity-induced expression of the first cluster of *Bdnf* transcripts. Furthermore, repressing or activating the +3 kb enhancer region with CRISPRi or CRISPRa, respectively, also affected the stimulus-induced levels of these transcripts. Notably, the part of the *Bdnf* gene containing the +3 kb enhancer has previously been implicated in the Reelin-mediated induction of *Bdnf* expression (*Telese et al., 2015*), indicating that the +3 kb enhancer could respond to other stimuli in addition to membrane depolarization and TrkB signaling.

Recently, it has been shown that other distal enhancer regions form chromatin loops with the *Bdnf* first cluster of exons and the +3 kb enhancer region (*Beagan et al., 2020*). Similarly, long-distance interactions with the +3 kb enhancer region have been reported in neurons isolated from the human brain (*Nott et al., 2019*). Furthermore, CTCF and RAD21, both important regulators of the chromatin 3D structure (*Rowley and Corces, 2018*), seem to bind near the +3 kb enhancer region (*Figure 8—figure supplement 1*). Therefore, it is possible that the +3 kb enhancer could partially mediate its effect on *Bdnf* gene expression by regulating the higher-order chromatin structure in the *Bdnf* gene locus by bringing together distal enhancer regions and *Bdnf* promoters from the first cluster of exons. Further work is necessary to determine the role of the +3 kb enhancer region in the regulation of higher order chromatin structure.

We also investigated the possibility that the +3 kb enhancer contributes to the catecholamine-induced expression of *Bdnf* transcripts in rat cultured cortical astrocytes (*Koppel et al., 2018*) and noted that even though the activation of the +3 kb enhancer increased the basal and stimulus-induced expression of all *Bdnf* transcripts, repression of the +3 kb enhancer had almost no effect on *Bdnf* expression. Furthermore, the transcriptional activity of the +3 kb enhancer was not induced by dopamine treatment in luciferase reporter assay, further indicating that the +3 kb region is not the enhancer responsible for catecholamine-dependent induction of *Bdnf* expression. Interestingly, the dopamine-dependent induction of *Bdnf* exon IIc-containing transcripts was abolished when the +3 kb enhancer was repressed using CRISPRi, suggesting that the +3 kb enhancer region might control the activity of stimulus-specific expression of *Bdnf* promoter II in astrocytes. Since the activity of *Bdnf* promoter II is regulated by neuron-restrictive silencer factor (NRSF) (*Timmusk et al., 1999*), it is possible that the drastic decrease in the dopamine-dependent induction of *Bdnf* exon IIc-containing transcripts was due to the cooperative effect between NRSF and the +3 kb enhancer region. Further investigation is needed to determine whether this hypothesis is true and whether such cooperation between the +3 kb enhancer region and NRSF binding to *Bdnf* exon II also happens in neurons. Although we have not tested it directly, our data does not support the notion that the +3 kb region is an active repressor in non-neuronal cells, for example, astrocytes. Instead, it seems that the +3 kb enhancer is a positive regulator of *Bdnf* gene operating specifically in neurons. We conclude that the +3 kb enhancer region is largely inactive in rat cultured cortical astrocytes and it is distinct from the distal *cis*-regulatory region controlling the catecholamine-induced activities of *Bdnf* promoters IV and VI.

Our results indicate that the +3 kb enhancer can receive regulatory inputs from various basic helix–loop–helix transcription factors, including TCF4 and its pro-neural heterodimerization partners NeuroD2 and NeuroD6. Single-cell RNA-seq analysis in the mouse cortex and hippocampus has indicated that mRNAs of NeuroD transcription factors are expressed mainly in excitatory neurons, similar to *Bdnf* (*Tasic et al., 2016*). It has been reported that NeuroD2 preferentially binds to E-boxes CAGCTG or CAGATG (*Fong et al., 2012*), which is in agreement with the functional E-box CAGATG sequence found in the +3 kb enhancer. Furthermore, it has been previously shown that NeuroD2 knock-out animals exhibit decreased *Bdnf* mRNA and protein levels in the cerebellum (*Olson et al., 2001*). However, Olson et al. found no change in *Bdnf* levels in the cerebral cortex of these knock-out animals. It is possible that different NeuroD family transcription factors regulate *Bdnf* expression in different brain areas and developmental stages, or that a compensatory mechanism between NeuroD2 and NeuroD6, both binding the +3 kb enhancer region in our in vitro DNA pulldown assay,

exists in cortical neurons of NeuroD2 knock-out animals. It has been well described that NeuroD transcription factors regulate neuronal differentiation (*Massari and Murre, 2000*), axonogenesis (*Bormuth et al., 2013*), neuronal migration (*Guzelsoy et al., 2019*), and proper synapse formation (*Ince-Dunn et al., 2006*; *Wilke et al., 2012*). As BDNF also has a role in the aforementioned processes (*Park and Poo, 2013*), it is plausible that at least some of the effects carried out by NeuroD family result from increasing *Bdnf* expression. Further work is needed to clarify the exact role of TCF4 and NeuroD transcription factors in *Bdnf* expression.

In conclusion, we have identified a novel intronic enhancer region governing the expression of neuron-specific *Bdnf* transcripts starting from the first cluster of exons – exons I, II, and III – in mammals. Exciting questions for further work are whether the +3 kb enhancer region is active in all neurons or in specific neuronal subtypes, and whether the activity of this enhancer element underlies in vivo contributions of BDNF to brain development and function.

## Materials and methods

### Cultures of rat primary cortical neurons

All animal procedures were performed in compliance with the local ethics committee. Cultures of cortical neurons from embryonic day (E) 21 Sprague Dawley rat embryos of both sexes were prepared as described previously (*Esvald et al., 2020*). The cells were grown in Neurobasal A (NBA) medium (Gibco, Waltham, MA) containing 1× B27 supplement (Gibco, Waltham, MA), 1 mM L-glutamine (Gibco, Waltham, MA), 100 U/ml penicillin, and 0.1 mg/ml streptomycin (Gibco, Waltham, MA) or 100 µg/ml Primocin (Invivogen, San Diego, CA) instead of penicillin/streptomycin at 37°C in 5% $CO_2$ environment. At 2 days in vitro (DIV), half of the medium was replaced with fresh supplemented NBA or the whole medium was replaced for cells transduced with lentiviruses. To inhibit the proliferation of non-neuronal cells, a mitotic inhibitor 5-fluoro-2′-deoxyuridine (final concentration 10 µM, Sigma-Aldrich, Saint Louis, MO) was added with the change of the medium.

### Cultures of rat primary cortical astrocytes

Cultures of cortical astrocytes were prepared from E21 Sprague Dawley rat embryos of both sexes as described previously (*Koppel et al., 2018*). The cells were grown in 75 cm² tissue culture flasks in Dulbecco's Modified Eagle Medium (DMEM with high glucose, PAN Biotech, Aidenbach, Germany) supplemented with 10% fetal bovine serum (PAN Biotech, Aidenbach, Germany) and 100 U/ml penicillin and 0.1 mg/ml streptomycin (Gibco, Waltham, MA) at 37°C in 5% $CO_2$ environment. At 1 DIV, the medium was replaced with fresh growth medium to remove loose tissue clumps. At 6 DIV, the flasks were placed into a temperature-controlled shaker Certomat BS-1 (Sartorius Group, Goettingen, Germany) for 17–20 hr and shaken at 180 rpm at 37°C to detach non-astroglial cells from the flask. After overnight shaking, the medium was removed along with unattached non-astrocytic cells, and astrocytes were washed three times with 1× phosphate-buffered saline (PBS). Astrocytes were detached from the flask with trypsin-EDTA solution (0.25% Trypsin-EDTA [1×], Gibco, Waltham, MA) diluted four times with 1× PBS at 37°C for 3–5 min. Trypsinized astrocytes were collected in supplemented DMEM and centrifuged at 200 × *g* for 6 min. The supernatant was removed, astrocytes were resuspended in supplemented DMEM and seeded on cell culture plates previously coated with 0.2 mg/ml poly-L-lysine (Sigma-Aldrich, Saint Louis, MO) in Milli-Q. At 9 DIV, the whole medium was replaced with fresh supplemented DMEM.

### Drug treatments

At 7 DIV, cultured neurons were pre-treated with 1 µM tetrodotoxin (Tocris, Bristol, UK) until the end of the experiment to inhibit spontaneous neuronal activity. At 8 DIV, neurons were treated with 50 ng/ml human recombinant BDNF (Peprotech, London, UK) or with a mixture of 25 mM KCl and 5 µM N-Methyl-D-aspartate receptor antagonist D-2-amino-5-phosphopentanoic acid (D-APV, Cayman Chemical Company, Ann Arbor, MI) to study BDNF autoregulation or neuronal activity-dependent expression of the *Bdnf* gene, respectively.

Cultured cortical astrocytes were treated at 15 DIV with 150 µM dopamine (Tocris, Bristol, UK) to study the regulation of the *Bdnf* gene by catecholamines or 0.15% dimethyl sulfoxide (Sigma-

Aldrich, Saint Louis, MO) as a vehicle control in fresh serum-free and antibiotics-free DMEM (DMEM with high glucose, PAN Biotech, Aidenbach, Germany).

## Transfection of cultured cells and luciferase reporter assay

Rat +3 kb enhancer (chr3:100771267–100772697, rn6 genome assembly) or +11 kb intron (chr3:100778398–100779836, rn6 genome assembly) regions were amplified from rat *Bdnf* BAC construct (*Koppel et al., 2018*) using Phusion Hot Start II DNA Polymerase (Thermo Fisher Scientific, Waltham, MA) and cloned into pGL4.15 vector (Promega, Madison, WI) in front of the Firefly luciferase coding sequence. To generate reporter constructs containing both *Bdnf* promoter and enhancer region, the hygromycin expression cassette downstream of firefly luciferase expression cassette in pGL4.15 vector was replaced with a new multiple cloning site, into which the +3 kb enhancer or +11 kb intron regions were cloned in either forward or reverse orientation (respective to the rat *Bdnf* gene). The *Bdnf* promoter regions were obtained from rat *Bdnf* promoter constructs (*Esvald et al., 2020*) and cloned in front of the firefly luciferase coding sequence. Plasmids encoding control and TCF4 shRNA have been published previously (*Sepp et al., 2017*). Coding regions of different dominant negative transcription factors were subcloned from AAV plasmids (*Esvald et al., 2020*) into pRRL vector backbone under the control of human *PGK* promoter.

For transfection and luciferase reporter assays, rat cortical neurons or astrocytes were grown on 48-well cell culture plates. Transfections were carried out in duplicate wells.

Cultured cortical neurons were transfected as described previously (*Jaanson et al., 2019*) with minor modifications. Transfection was carried out in unsupplemented NBA using 500 ng of the luciferase reporter construct and 20 ng of a normalizer plasmid pGL4.83-mPGK-hRLuc at 5–6 DIV using Lipofectamine 2000 (Thermo Scientific) with DNA to Lipofectamine ratio of 1:2. Transfection was terminated by replacing the medium with conditioned medium, which was collected from the cells before transfection.

Cultured cortical astrocytes were transfected as described previously (*Koppel et al., 2018*) using 190 ng of luciferase reporter construct and 10 ng of normalizer plasmid pGL4.83-SRα-hRLuc at 13 DIV using Lipofectamine 2000 (Thermo Scientific) with DNA to Lipofectamine ratio of 1:3.

The cells were lysed with Passive Lysis Buffer (Promega, Madison, WI) and luciferase signals were measured with Dual-Glo Luciferase assay kit (Promega, Madison, WI) using GENios pro plate reader (Tecan). Background-corrected firefly luciferase signals were normalized to background-corrected Renilla luciferase signals, and the averages of duplicate wells were calculated. Data were log-transformed for statistical analysis, mean and standard error of the mean (SEM) were calculated, and data were back-transformed for graphical representation.

## CRISPR interference and activator systems, RT-qPCR

pLV-hUbC-dCas9-KRAB-T2A-GFP plasmid used for CRISPR interference has been described previously (*Esvald et al., 2020*), and pLV-hUbC-VP64-dCas9-VP64-T2A-GFP plasmid used for CRISPR activation was obtained from Addgene (plasmid #59791). Lentiviral particles were produced as described previously (*Koppel et al., 2018*). Relative viral titers were estimated from provirus incorporation rate measured by qPCR, and equal amounts of functional viral particles were used for transduction in the following experiments. The efficiency of viral transduction was at least 90–95% based on EGFP expression in transduced cells.

Rat cortical neurons were transduced at 0 DIV, whereas cortical astrocytes were transduced after sub-culturing at 7 DIV. After treatments at 8 DIV for neurons or at 14 DIV for astrocytes, the cells were lysed and RNA was extracted with RNeasy Mini Kit (Qiagen, Hilden, Germany) using on-column DNA removal with RNase-Free DNase Set (Qiagen, Hilden, Germany). RNA concentration was measured with BioSpec-nano spectrophotometer (Shimadzu Biotech, Kyoto, Japan). cDNA was synthesized from equal amounts of RNA with Superscript III or Superscript IV reverse transcriptase (Invitrogen, Waltham, MA, USA) using oligo(dT)$_{20}$ or a mixture of oligo(dT)$_{20}$ and random hexamer primer (ratio 1:1, Microsynth, Balgach, Switzerland). To measure +3 kb enhancer eRNAs, cDNA was synthesized using a mixture of antisense eRNA-specific primer and *Hprt1* primer (1:1 ratio). The primers used for cDNA synthesis are listed in *Supplementary file 1*.

All qPCR reactions were performed in 10 µl volume in triplicates with 1× HOT FIREpol EvaGreen qPCR Mix Plus (Solis Biodyne, Tartu, Estonia) and primers listed in *Supplementary file 1* on

LightCycler 480 PCR instrument II (Roche, Basel, Switzerland). Gene expression levels were normalized to *Hprt1* mRNA levels in neurons and cyclophilin B mRNA levels in astrocytes. Data were log-transformed and autoscaled (as described in *Vandesompele et al., 2002*) for statistical analysis, mean and SEM were calculated, and data were back-transformed for graphical representation.

## Bioinformatic analysis

RNA-seq and ATAC-seq data from *Carullo et al., 2020* was obtained from Sequence Read Archive (accession numbers SRP261474 and SRP261642) and processed as described in *Sirp et al., 2020* with minor modifications. Trimming was performed with BBDuk using the following parameters: ktrim=r k=23 mink=11 hdist=1 tbo qtrim=lr trimq=10 minlen=50.

For RNA-seq data, STAR aligner (version 2.7.3a, *Dobin et al., 2013*) was used to map the reads to the Rn6 genome. Featurecounts (version 2.0.0, *Liao et al., 2014*) was used to assign reads to genes (annotation from Ensembl, release 101), and a custom-made SAF file was used to count reads mapping to +3 kb antisense eRNA (coordinates chr3:100771209–100772077). Counts were normalized using DESeq2 R package (version 1.28.1, *Love et al., 2014*) and visualized using ggplot2 R package (version 3.3.2, *Wickham, 2016*).

For ATAC-seq data, reads were mapped to the Rn6 genome with Bowtie2 (version 2.3.5.1, *Langmead and Salzberg, 2012*) using the following parameters: `--local --very-sensitive-local -X 3000`. The resultant SAM files were converted to BAM, sorted and indexed using Samtools (version 1.9, *Li et al., 2009*).

For both RNA-seq and ATAC-seq, aligned reads of biological replicates were merged using Samtools, converted to bigWig format using bamCoverage (version 3.3.0, *Ramírez et al., 2016*) with CPM normalization, and visualized using Integrative Genomics Viewer (*Robinson et al., 2011*).

## Mouse embryonic stem cells

A2Lox mESCs were a kind gift from Michael Kyba. The cells were authenticated based on their ability to form stable G418-resistant colonies after transfecting doxycycline-treated cultures with the p2Lox plasmid (Addgene plasmid #34635, *Iacovino et al., 2011*). The cells were tested negative for mycoplasma using PCR Mycoplasma Test Kit I/C (PromoCell, Heidelberg, Germany). A2Lox mESC line containing doxycycline-inducible Neurogenin2 transgene (Zhuravskaya et al., in preparation) was generated using recombination-mediated cassette exchange procedure (*Iacovino et al., 2011*). mESCs were grown in 2i media as described in *Iacovino et al., 2011* and *Kainov and Makeyev, 2020*. To delete the +3 kb enhancer region, 3 + 3 gRNAs targeting either side of the +3 kb enhancer core region (targeting sequences listed in *Supplementary file 2*) were cloned into pX330 vector (Addgene plasmid #42230). A2Lox-Neurogenin2 mESCs were co-transfected with a mixture of all 6 CRISPR plasmids and a plasmid containing a blasticidin expression cassette for selection. One day post transfection, 8 µg/ml blasticidin (Sigma-Aldrich, Saint Louis, MO) was added to the media for 3 days after which selection was ended, and cells were grown for an additional 11 days. Finally, single colonies were picked and passaged. The deletion of the +3 kb enhancer region was assessed from genomic DNA with PCR using primers flanking the desired deletion area. To rule out larger genomic deletions, qPCR-based copy number analysis was carried out with primers targeting the desired deletion area, and either side of the +3 kb region outside of the desired deletion area. qPCR results were normalized to the levels of the *Sox2* genomic locus, and the copy number of each region in wild-type cells was set as 2. Normalized copy numbers in recombinant clones were rounded to the nearest integer. All primers used for genotyping are listed in *Supplementary file 1*. Cell clones containing no deletion, heterozygous or homozygous deletion of the core conserved enhancer region, together with intact distal flanking regions, were used for subsequent analysis.

Selected mESC clones were differentiated into neurons as follows. Cells were plated on 12-well plates coated with Matrigel (Gibco, Waltham, MA) at a density of ~25,000 cells per well in N2B27 media (1:1 DMEM F12-HAM and Neurobasal mixture, 1× N2, 1× B27 with retinoic acid, 1× penicillin-streptomycin, 1 µg/ml laminin, 20 µg/ml insulin, 50 µM L-glutamine) supplemented with 0.1 M β-mercaptoethanol (Sigma-Aldrich, Saint Louis, MO) and 2 µg/ml doxycycline (Sigma-Aldrich, Saint Louis, MO). After 2 days, the whole media was changed to N2B27 media containing 200 µM ascorbic acid (Sigma-Aldrich, Saint Louis, MO) and 1 µg/ml doxycycline. Next, half of the media was replaced every 2 days with new N2B27 media containing 200 µM ascorbic acid but no doxycycline.

On the 12th day of differentiation, cells were treated with Milli-Q, BDNF (Peprotech, London, UK), or KCl for 3 hr. All treatments were added together with 25 µM D-APV (Alfa Aesar, Kandel, Germany). After treatment, the cells were lysed and RNA was extracted using EZ-10 DNAaway RNA Mini-Prep Kit (Bio Basic inc, Markham, Canada). cDNA was synthesized using Superscript IV (Thermo Fischer, Waltham, MA), and qPCR was performed with HOT FIREPol EvaGreen qPCR Mix Plus (Solis Biodyne, Tartu, Estonia) or qPCRBIO SyGreen Mix Lo-ROX (PCR Biosystems Ltd, London, UK) on LightCycler 96 (Roche, Basel, Switzerland). The levels of *Cnot4* mRNA expression were used for normalization. RT-qPCR was also used to characterize differentiation of A2Lox-Neurogenin2 parental cells into neurons (*Figure 7—figure supplement 2*). All primers used in the RT-qPCR are listed in *Supplementary file 1*.

## In vitro DNA pulldown mass spectrometry

The 855 bp region of the +3 kb enhancer and +11 kb intronic region were amplified with PCR using HotFirePol polymerase (Solis Biodyne, Tartu, Estonia) and primers listed in *Supplementary file 1*, with the reverse primers having a 5' biotin modification (Microsynth, Balgach, Switzerland). PCR products were purified using DNA Clean and Concentrator−100 kit (Zymo Research, Irvine, CA) using a 1:5 ratio of PCR solution and DNA binding buffer. The concentration of the DNA was determined with Nanodrop 2000 spectrophotometer (Thermo Scientific).

The preparation of nuclear lysates was performed as follows. Cortices from 8-day-old Sprague Dawley rat pups of both sexes were dissected and snap-frozen in liquid nitrogen. Nuclear lysates were prepared with high salt extraction as in *Wu, 2006* and *Lahiri and Ge, 2000* with minor modifications. Briefly, cortices were weighed and transferred to pre-cooled Dounce tissue grinder (Wheaton). Also, 2 ml of ice-cold cytoplasmic lysis buffer (10 mM HEPES, pH 7.9 [adjusted with NaOH], 10 mM KCl, 1.5 mM $MgCl_2$, 0.5% NP-40, 300 mM sucrose, 1x cOmplete Protease Inhibitor Cocktail [Roche, Basel, Switzerland], and phosphatase inhibitors as follows: 5 mM NaF [Fisher Chemical, Pittsburgh, PA], 1 mM beta-glycerophosphate [Acros Organics, Pittsburgh, PA], 1 mM $Na_3VO_4$ [ChemCruz, Dallas,TX], and 1 mM $Na_4P_2O_7$ [Fisher Chemical, Pittsburgh, PA]) was added, and tissue was homogenized 10 times with tight pestle. Next, the lysate was transferred to a 15 ml tube and cytoplasmic lysis buffer was added to a total volume of 1 ml per 0.1 g of tissue. The lysate was incubated on ice for 10 min with occasional inverting. Next, the lysate was transferred to a 100 µm nylon cell strainer (VWR) to remove tissue debris and the flow-through was centrifuged at 2600 × $g$ at 4°C for 1 min to pellet nuclei. The supernatant (cytoplasmic fraction) was discarded and the nuclear pellet was resuspended in 1 ml per 1 g of tissue ice-cold nuclear lysis buffer (20 mM HEPES, pH 7.9, 420 mM NaCl, 1.5 mM $MgCl_2$, 0.1 mM EDTA, 2.5% glycerol, 1x cOmplete Protease Inhibitor Cocktail [Roche, Basel, Switzerland], and phosphatase inhibitors) and transferred to a new Eppendorf tube. To extract nuclear proteins, the pellet was rotated at 4°C for 30 min and finally centrifuged at 11,000 × $g$ at 4°C for 10 min. The supernatant was collected as nuclear fraction, and protein concentration was measured with BCA Protein Assay Kit (Pierce).

In vitro DNA pulldown was performed as follows. Two biological replicates were performed using nuclear lysates of cortices from pups of different litters. Pierce Streptavidin Magnetic Beads (50 µl per pulldown reaction) were washed 2 times with 1× binding buffer (BB, 5 mM Tris-HCl, pH 7.5, 0.5 mM EDTA, 1 M NaCl, 0.05% Tween-20), resuspended in 2× BB, and an equal volume of 50 pmol biotinylated DNA (in 10 mM Tris-HCl, pH 8.5, 0.1 mM EDTA) was added and incubated at room temperature for 30 min with rotation. To remove the unbound probe, the beads were washed three times with 1× BB. Finally, 400 µg of nuclear proteins (adjusted to a concentration of 1.6 mg/ml with nuclear lysis buffer) and an equal volume of buffer D (20 mM HEPES, pH 7.9, 100 mM KCl, 0.2 mM EDTA, 8% glycerol, 1x cOmplete Protease Inhibitor Cocktail [Roche, Basel, Switzerland], and phosphatase inhibitors) were added and incubated with rotation at 4°C overnight. The next day the beads were washed three times with 1× PBS, once with 100 mM NaCl and once with 200 mM NaCl. Bound DNA and proteins were eluted with 16 mM biotin (Sigma-Aldrich, Saint Louis, MO) in water (at pH 7.0) at 80°C for 5 min, and the eluate was transferred to a new tube and snap-frozen in liquid nitrogen.

Mass-spectrometric analysis of the eluates was performed with nano-LC-MS/MS using Q Exactive Plus (Thermo Scientific) at Proteomics core facility at the University of Tartu, Estonia, as described previously (*Mutso et al., 2018*) using label-free quantification instead of SILAC and *Rattus norvegicus* reference proteome for analysis. The full lists of proteins obtained from mass-spectrometric

analysis are shown in *Supplementary file 3*. Custom R script (available at *Tuvikene, 2021*) was used to keep only transcription factors based on gene symbols of mammalian genes from gene ontology categories 'RNA polymerase II cis-regulatory region sequence-specific DNA binding' and 'DNA-binding transcription factor activity' from http://geneontology.org/ (obtained March 16, 2020). At least 1.45-fold enrichment to the +3 kb enhancer probe compared to the +11 kb intronic probe was used as a cutoff for specific binding. The obtained lists were manually curated to generate Venn diagram illustration of the experiment.

### Chromatin immunoprecipitation

ChIP assay was performed as described previously (*Esvald et al., 2020*) using 10 min fixation with 1% formaldehyde. 5 µg of CREB antibody (catalog #06–863, lot 2446851, Merck Millipore, Burlington, MA) or TCF4 antibody (CeMines, Golden, CO) was used per immunoprecipitation (IP). DNA enrichment was measured using qPCR. All qPCR reactions were performed in 10 µl volume in triplicates with 1× LightCycler 480 SYBR Green I Master kit (Roche, Basel, Switzerland) and primers listed in *Supplementary file 1* on LightCycler 480 PCR instrument II (Roche, Basel, Switzerland). Primer efficiencies were determined by serial dilutions of input samples and were used for analyzing the results. Percentage of input enrichments was calculated for each region and IP, and data were log-transformed before statistical analysis.

ENCODE data of different ChIP-seq experiments were visualized using UCSC Genome Browser track 'Transcription Factor ChIP-seq Peaks (340 factors in 129 cell types) from ENCODE 3 Data version: ENCODE 3 Nov 2018'. Data of previously published ChIP-seq experiments were obtained from Gene Expression Omnibus with accession numbers GSM530173, GSM530174, GSM530182, GSM530183 (*Kim et al., 2010*), GSM1467429, GSM1467434 (*Malik et al., 2014*), GSM1820990 (*Moen et al., 2017*), GSM1647867 (*Sun et al., 2019*), GSM1649148 (*Bayam et al., 2015*), and visualized using Integrative Genomics Viewer version 2.8.0 (*Robinson et al., 2011*).

### Statistical analysis

Sample size estimation was not performed, and randomization and blinding were not used. Statistical analysis was performed on data obtained from independent biological replicates – cultured primary cells obtained from pups of different litters, or different clones of mESCs. All statistical tests and tested hypotheses were decided before performing the experiments. As ANOVA's requirement of homoscedasticity was not met, two-tailed paired or unpaired equal variance t-test, as reported at each figure, was used for statistical analysis using Excel 365 (Microsoft). To preserve statistical power, p-values were not corrected for multiple comparisons as recommended by *Feise, 2002*, *Rothman, 1990*, and *Streiner and Norman, 2011*.

## Acknowledgements

This work was supported by Estonian Research Council (institutional research funding IUT19-18 and grant PRG805), Norwegian Financial Mechanism (Grant EMP128), European Union through the European Regional Development Fund (Project No. 2014–2020.4.01.15-0012), H2020-MSCA-RISE-2016 (Grant EU734791), and the Biotechnology and Biological Sciences Research Council (BB/M001199/1, BB/M007103/1, and BB/R001049/1). This work has also been partially supported by 'TUT Institutional Development Program for 2016–2022' Graduate School in Clinical medicine receiving funding from the European Regional Development Fund under program ASTRA 2014–2020.4.01.16-0032 in Estonia. We thank Epp Väli and Andra Moistus for technical assistance, and Indrek Koppel and Priit Pruunsild for critical reading of the manuscript.

## Additional information

### Competing interests

Jürgen Tuvikene, Eli-Eelika Esvald, Annika Rähni, Tõnis Timmusk: is an employee of Protobios LLC. The other authors declare that no competing interests exist.

## Funding

| Funder | Grant reference number | Author |
|---|---|---|
| Estonian Research Council | IUT19-18 | Jürgen Tuvikene<br>Eli-Eelika Esvald<br>Annika Rähni<br>Kaie Uustalu<br>Annela Avarlaid<br>Tõnis Timmusk |
| Estonian Research Council | PRG805 | Jürgen Tuvikene<br>Eli-Eelika Esvald<br>Annela Avarlaid<br>Tõnis Timmusk |
| Norwegian Financial Mechanism | EMP128 | Jürgen Tuvikene<br>Eli-Eelika Esvald<br>Annika Rähni<br>Kaie Uustalu<br>Tõnis Timmusk |
| European Regional Development Fund | 2014-2020.4.01.15-0012 | Jürgen Tuvikene<br>Eli-Eelika Esvald<br>Annika Rähni<br>Kaie Uustalu<br>Annela Avarlaid<br>Tõnis Timmusk |
| H2020 Marie Skłodowska-Curie Actions | EU734791 | Jürgen Tuvikene<br>Eli-Eelika Esvald<br>Anna Zhuravskaya<br>Annela Avarlaid<br>Eugene V Makeyev<br>Tõnis Timmusk |
| Biotechnology and Biological Sciences Research Council | BB/M001199/1 | Anna Zhuravskaya<br>Eugene V Makeyev |
| Biotechnology and Biological Sciences Research Council | BB/M007103/1 | Anna Zhuravskaya<br>Eugene V Makeyev |
| Biotechnology and Biological Sciences Research Council | BB/R001049/1 | Anna Zhuravskaya<br>Eugene V Makeyev |
| European Regional Development Fund | ASTRA 2014-2020.4.01.16-0032 | Jürgen Tuvikene<br>Eli-Eelika Esvald<br>Annela Avarlaid<br>Tõnis Timmusk |

The funders had no role in study design, data collection and interpretation, or the decision to submit the work for publication.

## Author contributions

Jürgen Tuvikene, Conceptualization, Data curation, Formal analysis, Supervision, Investigation, Visualization, Methodology, Writing - original draft, Writing - review and editing; Eli-Eelika Esvald, Conceptualization, Formal analysis, Investigation, Methodology, Writing - review and editing; Annika Rähni, Formal analysis, Investigation, Methodology, Writing - review and editing; Kaie Uustalu, Formal analysis, Investigation, Methodology; Anna Zhuravskaya, Supervision, Investigation, Methodology, Writing - review and editing; Annela Avarlaid, Formal analysis, Supervision, Investigation, Methodology, Writing - review and editing; Eugene V Makeyev, Conceptualization, Supervision, Funding acquisition, Methodology, Writing - review and editing; Tõnis Timmusk, Conceptualization, Supervision, Funding acquisition, Project administration, Writing - review and editing

## Author ORCIDs

Jürgen Tuvikene https://orcid.org/0000-0002-9665-760X
Eli-Eelika Esvald https://orcid.org/0000-0001-5730-8207
Annika Rähni https://orcid.org/0000-0002-2826-4636
Anna Zhuravskaya https://orcid.org/0000-0001-9312-1212
Annela Avarlaid https://orcid.org/0000-0002-5941-8271

Eugene V Makeyev (iD) https://orcid.org/0000-0001-6034-6896
Tõnis Timmusk (iD) https://orcid.org/0000-0002-1015-3348

**Decision letter and Author response**
Decision letter https://doi.org/10.7554/eLife.65161.sa1
Author response https://doi.org/10.7554/eLife.65161.sa2

## Additional files

### Supplementary files

• Supplementary file 1. List of used primers.

• Supplementary file 2. List of gRNA targeting sequences.

• Supplementary file 3. Mass-spectrometry results of the in vitro DNA pulldown experiment. Max-Quant ProteinGroups tables of two biological replicates are shown in separate Excel sheets. Refer to the MaxQuant documentation (http://coxdocs.org/doku.php?id=maxquant:table:proteingrouptable) for detailed column descriptions.

• Transparent reporting form

### Data availability

Mass-spectrometry results of the in vitro DNA pulldown experiment are provided in Supplementary file 3.

The following previously published datasets were used:

| Author(s) | Year | Dataset title | Dataset URL | Database and Identifier |
|---|---|---|---|---|
| Kim T, Hemberg M, Gray JM, Kreiman G, Greenberg ME | 2010 | Widespread transcription at neuronal activity-regulated enhancers | https://www.ncbi.nlm.nih.gov/geo/query/acc.cgi?acc=GSE21161 | NCBI Gene Expression Omnibus, GSE21161 |
| Nord AS, Visel A, Pennacchio LA | 2013 | Rapid and Pervasive Changes in Genome-Wide Enhancer Usage During Mammalian Development | https://www.ncbi.nlm.nih.gov/geo/query/acc.cgi?acc=GSE52386 | NCBI Gene Expression Omnibus, GSE52386 |
| Malik AN | 2014 | Genome-wide identification and characterization of functional neuronal activity-dependent enhancers | https://www.ncbi.nlm.nih.gov/geo/query/acc.cgi?acc=GSE60192 | NCBI Gene Expression Omnibus, GSE60192 |
| Dunn GI | 2015 | NeuroD2 ChIP-SEQ from embryonic cortex | https://www.ncbi.nlm.nih.gov/geo/query/acc.cgi?acc=GSE67539 | NCBI Gene Expression Omnibus, GSE67539 |
| Brandsma JH, Moen MJ, Poot RA | 2016 | A protein interaction network of mental disorder factors in neural stem cells | https://www.ncbi.nlm.nih.gov/geo/query/acc.cgi?acc=GSE70872 | NCBI Gene Expression Omnibus, GSE70872 |
| Sun Z, Sun M, He J, Xie H | 2019 | Genome-wide maps of EGR1 binding in mouse frontal cortex | https://www.ncbi.nlm.nih.gov/geo/query/acc.cgi?acc=GSE67482 | NCBI Gene Expression Omnibus, GSE67482 |
| Carullo NVN, Phillips RA, Ianov L, Day JJ | 2020 | ATAC-seq datasets for chromatin accessibility quantification in "Enhancer RNAs predict enhancer-gene regulatory links and are critical for enhancer function in neuronal systems" | https://www.ncbi.nlm.nih.gov/geo/query/acc.cgi?acc=GSE150589 | NCBI Gene Expression Omnibus, GSE150589 |
| Carullo NVN, Phillips RA, Ianov L, Day JJ | 2020 | RNA-seq datasets for enhancer RNA quantification in "Enhancer RNAs predict enhancer-gene regulatory links and are critical for enhancer function in neuronal systems" | https://www.ncbi.nlm.nih.gov/geo/query/acc.cgi?acc=GSE150499 | NCBI Gene Expression Omnibus, GSE150499 |

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
