## [Decision Letter]

**Acceptance summary:**

This paper presents a comprehensive evaluation of an intragenic transcriptional enhancer element that promotes transcription of the gene encoding Brain-Derived Neurotrophic Factor (BDNF). This is the first enhancer to be validated for this important activity-regulated neuronal plasticity and development gene. In addition to demonstrating the function of the enhancer in the specificity of BDNF promoter activation in both neurons and astrocytes, the authors also identify transcriptional regulators of enhancer activity.

**Decision letter after peer review:**

Thank you for submitting your article "Intronic enhancer region governs transcript-specific BDNF expression in neurons" for consideration by *eLife*. Your article has been reviewed by three peer reviewers, one of whom is a member of our Board of Reviewing Editors, and the evaluation has been overseen by Gary Westbrook as the Senior Editor. The following individual involved in review of your submission has agreed to reveal their identity: Svitlana Bach (Reviewer #3). The reviewers have discussed the reviews with one another and the Reviewing Editor has drafted this decision to help you prepare a revised submission.

We would like to draw your attention to changes in our policy on revisions we have made in response to COVID-19 (https://elifesciences.org/articles/57162). Specifically, when editors judge that a submitted work as a whole belongs in *eLife* but that some conclusions require a modest amount of additional new data, as they do with your paper, we are asking that the manuscript be revised to either limit claims to those supported by data in hand, or to explicitly state that the relevant conclusions require additional supporting data.

Summary:

In this manuscript, the authors set out to find distal enhancers that regulate BDNF, a biologically important gene whose dynamic expression is primarily controlled at the transcriptional level. Enhancers are well known to play key roles in the fine tuning of gene regulation as well as in the control of cell-type specificity. However, it is surprising that to date, the majority of data regarding mechanisms of BDNF transcription have focused on promoters rather than enhancers. Here the authors bioinformatically identify a putative BDNF enhancer and then validate its function and regulation using luciferase assays, dCas9-mediated functional genomics, and CRISPR knockout, followed by the identification of transcriptional regulators of this element. This is a comprehensive and important validation of a new regulatory element that contributes to regulation of the BDNF gene. The characterization of the element defined here will allow others in the field to determine the importance of this regulatory mechanism for the control of *Bdnf* expression in the wide range of biological processes in which it is implicated.

Essential revisions:

The reviewers thought the conclusions were strongly supported by the data and have only three areas they think need to be strengthened with the addition of some details in the text.

1) With respect to experimental details the reviewers asked that the following information be added/clarified:

a) The stem cell lines need to be more fully described and the characterization shown or directly referenced. The authors stated there is a manuscript in preparation describing the Ngn2 line, but for the tools used here there must be sufficient information provided to evaluate them. This includes sequence confirmation of the targeting and validation of the differentiation at a minimum.

b) What is the efficacy of the viral vectors used to deliver the dCas9 fusions and gRNAs? This information is required for interpreting the efficacy of *Bdnf* gene induction. A 50% reduction could reflect full blockade in 50% of the cells or a partial block in all of the cells.

c) Could the authors clarify how the luciferase vectors used here work? Most studies clone the enhancer region of interest into vectors that have a weak core promoter (like the SV40 promoter in the pGL3 vector system, or the Hsp68 promoter used by the VISTA enhancer project) rather than into a vector with no promoter. Is this enhancer acting as a core promoter?

2) With respect to the text matching the statistics the reviewers ask the authors to double check the following:

a) "…had virtually no effect on the response of these promoters to dopamine treatment." However, Figure 3D and E depicts significance asterisks above the +3kb enh forward 8h DA bar graph.

b) "… +11 kb intronic region with dCas9 without an effector domain had no major effect…" There appears to be at least some dampening of *Bdnf* transcript upregulation after stimulation, perhaps as a result of CRISPR interference (Figure 4, left panel). For example, there is a significance asterisk above the +3kb enh gRNA after 3h KCl stimulation for *Bdnf* exon I mRNA expression.

c) "…CRISPRi decreased the basal expression levels of BDNF exon I, IIc, and III- containing transcripts by 2.2, 11, and 2.4-fold, respectively (Figure 4A, B, C, middle panel). In contrast, no significant effect was seen…" As shown in Figure 4, middle panel, expression levels of BDNF exon I, IIc, and III are not significantly downregulated.

d) "…targeting CRISPRa to the +11 kb intronic region had no significant effect on the expression of any of the BDNF transcripts…" This statement does not match the significance asterisks indicated in Figure 4A and E, right panel.

e) "Targeting dCas9 to the BDNF locus did not affect the expression of any of the BDNF transcripts." This does not match the significance asterisks in Figure 5B and E, left panel, which represent some CRISPRi effects.

f) "…targeting CRISPRi to the +3 kb enhancer region in astrocytes did not affect the basal expression levels of any BDNF transcript." This statement does not match the significance asterisk indicated in Figure 5D, middle panel.

g) The "no significant effect" statement in Figure 6 seems at odds with the corresponding data presented in Figures 4 and 5. Maybe the phrase can be changed to "no noteworthy effect."

3) Finally with respect to the Discussion the authors note that in addition to the papers cited (Kim et al., 2010, Malik et al., 2014) the enhancer of interest in this paper could also be seen bioinformatically in another recent publication that sought putative distal enhancers of BDNF using 5C (PMID# 32451484). This reference should be added and the authors might wish to discuss.

---

## [Author Response]

Essential revisions:The reviewers thought the conclusions were strongly supported by the data and have only three areas they think need to be strengthened with the addition of some details in the text.1) With respect to experimental details the reviewers asked that the following information be added/clarified:a) The stem cell lines need to be more fully described and the characterization shown or directly referenced. The authors stated there is a manuscript in preparation describing the Ngn2 line, but for the tools used here there must be sufficient information provided to evaluate them. This includes sequence confirmation of the targeting and validation of the differentiation at a minimum.

The parental A2Lox-Neurogenin2 mouse ESC line was generated using recombination-mediated cassette exchange procedure referenced in our manuscript (Iacovino et al., 2011). This is a routine procedure where correct recombinants are simply selected based on their G418 resistance. As requested, we have now analyzed the time course of doxycycline-induced differentiation of these G418-resistant cells using an RT-qPCR assay for classical stem cell, neural progenitor and neuronal markers (*Pou5F1*/OCT4, *Sox2*, *Nes*/nestin, *Map2*, *Tubb3*, and the AMPA receptor subunit *Gria1*; new Figure 7—figure supplement 2).

The clonal cell lines generated with CRISPR/Cas9 from A2Lox-Neurogenin2 contain different deletions of the +3 kb region. Although we have not sequenced the exact deletion boundaries, we estimated the approximate deletion size for all clones using PCR and agarose gel electrophoresis (data not shown) and performed qPCR-based locus copy number assays with primer pairs specific to the +3kb core region and its flanking sequences. We now show this copy number genotyping data in the new Figure 7—figure supplement 1.

b) What is the efficacy of the viral vectors used to deliver the dCas9 fusions and gRNAs? This information is required for interpreting the efficacy of Bdnf gene induction. A 50% reduction could reflect full blockade in 50% of the cells or a partial block in all of the cells.

The efficiency of viral transduction was at least 90-95%, as evident from EGFP expression in transduced cells. Therefore, the CRISPR/dCas9 system was present in most cells. This in turn means that our CRISPRi results should be interpreted as partial inhibition of the *Bdnf* gene expression in almost all cells.

c) Could the authors clarify how the luciferase vectors used here work? Most studies clone the enhancer region of interest into vectors that have a weak core promoter (like the SV40 promoter in the pGL3 vector system, or the Hsp68 promoter used by the VISTA enhancer project) rather than into a vector with no promoter. Is this enhancer acting as a core promoter?

We have used two types of luciferase reporter vectors in our study – ones where the enhancer region was cloned directly in front of the firefly coding sequence without an additional promoter region (enhancer-luciferase configuration), and others containing *Bdnf* promoter region in front of the luciferase coding sequence and enhancer region downstream of the luciferase expression cassette (promoter-luciferase-enhancer configuration).

It is well known that enhancer regions can initiate RNA polymerase II-dependent bidirectional transcription of enhancer RNAs. While endogenous eRNA transcripts are usually unstable, it seems that in heterologous context in the luciferase vectors using the enhancer-luciferase configuration the resulting RNAs are more stable, possibly due to polyadenylation and differences in chromatin structure of the region, and can therefore be translated to detectable amount of luciferase protein. Furthermore, considering that the +3 kb enhancer region itself shows very strong transcription (e.g. much stronger than *Bdnf* promoter I) when used as a promoter in reporter assay, cloning it upstream of a minimal promoter or *Bdnf* promoter I would generate both enhancer- and promoter-initiated transcripts with slightly different 5' UTRs. Therefore, the results obtained with luciferase vectors using the enhancer–promoter–luciferase configuration would be difficult to interpret. For this reason, to study the synergy between promoter and enhancer region, we have used luciferase vectors with the promoter-luciferase-enhancer configuration.

2) With respect to the text matching the statistics the reviewers ask the authors to double check the following:a) "…had virtually no effect on the response of these promoters to dopamine treatment." However, Figure 3D and E depicts significance asterisks above the +3kb enh forward 8h DA bar graph.b) "… +11 kb intronic region with dCas9 without an effector domain had no major effect…" There appears to be at least some dampening of Bdnf transcript upregulation after stimulation, perhaps as a result of CRISPR interference (Figure 4, left panel). For example, there is a significance asterisk above the +3kb enh gRNA after 3h KCl stimulation for Bdnf exon I mRNA expression.c) "…CRISPRi decreased the basal expression levels of BDNF exon I, IIc, and III- containing transcripts by 2.2, 11, and 2.4-fold, respectively (Figure 4A, B, C, middle panel). In contrast, no significant effect was seen…" As shown in Figure 4, middle panel, expression levels of BDNF exon I, IIc, and III are not significantly downregulated.d) "…targeting CRISPRa to the +11 kb intronic region had no significant effect on the expression of any of the BDNF transcripts…" This statement does not match the significance asterisks indicated in Figure 4A and E, right panel.e) "Targeting dCas9 to the BDNF locus did not affect the expression of any of the BDNF transcripts." This does not match the significance asterisks in Figure 5B and E, left panel, which represent some CRISPRi effects.f) "…targeting CRISPRi to the +3 kb enhancer region in astrocytes did not affect the basal expression levels of any BDNF transcript." This statement does not match the significance asterisk indicated in Figure 5D, middle panel.g) The "no significant effect" statement in Figure 6 seems at odds with the corresponding data presented in Figures 4 and 5. Maybe the phrase can be changed to "no noteworthy effect."

We thank the reviewers for this comment. We have double-checked the statistical analyses and found no errors in our calculations or depiction of the test results in the figures. While the results pointed out are statistically significant, the effect size and the context of the experiment need to be considered when interpreting the results. We find that these solitary statistically significant data points do not change the main conclusions of our study. However, we agree that using the word “significant” as a synonym of “major” or “marked” may be confusing. The revised text has been changed accordingly to avoid this problem.

3) Finally with respect to the Discussion the authors note that in addition to the papers cited (Kim et al., 2010, Malik et al., 2014) the enhancer of interest in this paper could also be seen bioinformatically in another recent publication that sought putative distal enhancers of BDNF using 5C (PMID# 32451484). This reference should be added and the authors might wish to discuss.

We thank the reviewers for the reference. However, to the best of our knowledge, this paper does not directly show that the +3 kb region is an enhancer region. Instead, it appears that other distal enhancer regions of the *Bdnf* gene use the +3 kb region (or the whole first cluster of *Bdnf* exons, it is difficult to say considering the resolution of the 5C) as a contact point for chromatin looping. Similar observation has been made by Nott et al., 2019, in neurons isolated from the human brain (PMID# 31727856). This notion raises the possibility that the +3 kb enhancer could partially mediate its effect on *Bdnf* expression by participating in regulating the higher-order chromatin structure at the *Bdnf* gene locus. We now mention this possibility in the revised Discussion section.